# Diffusion Auto-regressive Transformer for Effective Self-supervised Time Series Forecasting

## Abstract

Self-supervised learning has become a popular and effective approach for enhancing time series forecasting, enabling models to learn universal representations from unlabeled data. However, effectively capturing both the global sequence dependence and local detail features within time series data remains challenging. To address this, we propose a novel generative self-supervised method called **TimeDART**, denoting **D**iffusion **A**uto-regressive **T**ransformer for **T**ime series forecasting. In TimeDART, we treat time series patches as basic modeling units. Specifically, we employ an self-attention based Transformer encoder to model the dependencies of inter-patches. Additionally, we introduce diffusion and denoising mechanisms to capture the detail locality features of intra-patch. Notably, we design a cross-attention-based denoising decoder that allows for adjustable optimization difficulty in the self-supervised task, facilitating more effective self-supervised pre-training. Furthermore, the entire model is optimized in an auto-regressive manner to obtain transferable representations. Extensive experiments demonstrate that TimeDART achieves state-of-the-art fine-tuning performance compared to the most advanced competitive methods in forecasting tasks. Our code is publicly available[1].

## 1 Introduction

Time series forecasting (Harvey, 1990; Hamilton, 2020; Box et al., 2015) is crucial in a wide array of domains, including finance (Black & Scholes, 1973), healthcare (Cheng et al., 2024), energy management (Zhou et al., 2024). Accurate predictions of future data points could enable better decision-making, resource allocation, and risk management, ultimately leading to significant operational improvements and strategic advantages. Among the various methods developed for time series forecasting (Miller et al., 2024), deep neural networks (Ding et al., 2024; Jin et al., 2023; Cao et al., 2023) have emerged as a popular and effective solution paradigm.

To further enhance the performance of time series forecasting, self-supervised learning has become an increasingly popular research paradigm (Nie et al., 2022). This approach allows models to learn transferable representations from unlabeled data by self-supervised pre-training, which can then be fine-tuned for forecasting tasks. Scrutinizing previous studies (Zhang et al., 2024), existing methods primarily fall into two categories. The first category is masked autoencoders (Devlin, 2018; He et al., 2022), with representative methods including TST (Zerveas et al., 2021), TimeMAE (Cheng et al., 2023), and SimMTM (Dong et al., 2024). These methods focus on reconstructing masked or corrupted parts of the input data, encouraging the model to learn meaningful representations that capture the underlying structure of the time series. The second category comprises contrastive-based discriminative methods (Oord et al., 2018; Chen et al., 2020), such as TS-TCC (Tonekaboni et al., 2021), TS2Vec (Yue et al., 2022), and TNC (Eldele et al., 2021). These approaches leverage contrastive learning to distinguish between similar and dissimilar time series segments, thereby enhancing the model's ability to capture essential patterns and temporal dynamics.

Despite advancements in self-supervised methods, notable limitations persist when applying them to time series forecasting. First, masked methods introduce a significant gap between pre-training and

---

[1]https://anonymous.4open.science/r/TimeDART-2024

fine-tuning due to altered data distribution, which hinders effective representation transfer (Chen et al., 2024). Second, contrastive learning methods face challenges in constructing positive and negative pairs, given time series' temporal dependencies and ambiguity in defining similarity. These methods also prioritize learning discriminative features over modeling the generative aspects needed for forecasting (Cheng et al., 2023), limiting their ability to capture nuanced temporal dependencies.

Despite the recent advancements in self-supervised learning methods for time series (Zhang et al., 2024), we argue that an ideal approach should possess the following two key characteristics. First, the gap between the pre-training objective and the downstream fine-tuning task should be minimized as much as possible. As we know, the widely used one-step generation (Zhou et al., 2021) approach essentially employs an inductive bias of using the past to predict the future. In fact, auto-regressive generative optimization (Radford, 2018) aligns well with this paradigm (Liu et al., 2024; Liu et al.), yet it has rarely been adopted in the field of time series self-supervised learning. Second, it is crucial to model both long-term dependencies and local patterns during self-supervised pre-training of time series. However, existing self-supervised methods often struggle to effectively capture these aspects simultaneously, which significantly limits their ability to learn comprehensive and expressive representations of time series data. In this context, developing a novel approach that can effectively address the challenges discussed above is crucial to fully exploit the intricate temporal relationships present in time series.

Building upon this analysis above, in this work, we propose a novel self-supervised time series method called TimeDART. The key feature of TimeDART lies in its elegant integration of two advanced generative self-supervised approaches within a unified framework, allowing for effective self-supervised learning by simultaneously capturing both long-term dependencies and fine-grained local features in time series data. Specifically, we treat time series patches as the fundamental modeling units. To capture inter-patch dependencies, we employ a self-attention-based Transformer encoder. Concurrently, we introduce a forward diffusion and reverse denoising process to reconstruct the detailed features of individual patches, thereby effectively modeling local relational dependencies. Notably, within the diffusion module, we design a novel cross-attention-based denoising network that enables more flexible and adaptive noise reduction. Through this design, the TimeDART framework aims to shorten the gap between pre-training and fine-tuning tasks, while effectively modeling both global dependencies and local feature representations during the self-supervised learning process. Finally, we evaluate the effectiveness of our method on public datasets, demonstrating its superior performance over existing competitive approaches. We hope that TimeDART's strong performance can inspire more research work in this area.The main contribution of this work can be summarized as:

- We propose a novel generative self-supervised learning framework, TimeDART, which integrates diffusion and auto-regressive modeling to effectively learn both global sequence dependencies and local detail features from time series data, addressing the challenges of capturing comprehensive temporal characteristics.

- We design a cross-attention-based denoising decoder within the diffusion mechanism, which enables adjustable optimization difficulty during the self-supervised task. This design significantly enhances the model's ability to capture localized intra-patch features, improving the effectiveness of pre-training for time series forecasting.

- We conduct extensive experiments to validate that TimeDART achieves more superior performance on time series forecasting tasks. We also report some insight findings to understand the proposed TimeDART.

## 2 RELATED WORK

**Time Series Forecasting.** In recent years, deep learning-based models have significantly advanced time series forecasting by addressing long-range dependencies. Informer (Zhou et al., 2021) introduced ProbSparse attention to reduce complexity from $O(L^2)$ to $O(L \log L)$, combined with attention distillation to handle ultra-long inputs. Autoformer (Wu et al., 2022) proposed a decomposition architecture with an auto-correlation mechanism to improve efficiency and accuracy. FEDformer (Zhou et al., 2022) integrated seasonal-trend decomposition with frequency-enhanced attention, further reducing complexity to $O(L)$. Crossformer (Zhang & Yan, 2023) addressed multivariate time series forecasting by capturing both temporal and cross-dimensional dependencies through

dual-stage attention. PatchTST (Nie et al., 2022) introduced a patching strategy with channel independence and self-supervised pretraining, while iTransformer (Liu et al., 2023) applied attention and feedforward networks along reversed dimensions without altering the Transformer architecture. SimMTM (Dong et al., 2024) employed manifold learning to restore masked time points, improving semantic recovery, and GPHT (Liu et al., 2024) introduced a mixed dataset pretraining approach, enabling large-scale training and autoregressive forecasting without custom heads. Diffusion-TS (Yuan & Qiao, 2024) uses an encoder-decoder transformer to generate high-quality multivariate time series in a diffusion-based framework. These methods collectively enhance the efficiency, scalability, and accuracy of time series forecasting using Transformer architectures.

**Self-supervised Learning in Time Series.** Self-supervised learning has emerged as a powerful paradigm for pretraining in many domains, including natural language processing (NLP) and computer vision (CV). Unlike supervised learning, where models are trained with labeled data, self-supervised methods rely on the structure within the data itself to generate supervision, typically through pretext tasks. In the domain of time series, self-supervised learning faces unique challenges due to the sequential nature and temporal dependencies of the data. Current approaches can be broadly categorized into two paradigms: discriminative and generative methods.

Discriminative methods, such as contrastive learning, focus on distinguishing between positive and negative instance pairs. These methods learn representations by pulling similar instances (positive pairs) closer and pushing dissimilar instances (negative pairs) apart. For instance, TNC (Eldele et al., 2021) leverages the local smoothness of time series signals to define positive neighborhoods, while TS2Vec (Yue et al., 2022) introduces a hierarchical contrastive learning framework that operates at both the instance and patch levels. Similarly, CoST (Woo et al., 2022) incorporates both time and frequency domain information to capture seasonal and trend representations, improving the discriminative power of the learned features.

On the other hand, generative methods typically involve reconstructing masked or corrupted inputs, encouraging the model to learn meaningful representations. Masked time series modeling, first introduced by TST (Zerveas et al., 2021), predicts missing time points based on the available data. This approach has since been extended by methods like STEP (Shao et al., 2022) and PatchTST (Nie et al., 2022), which operate on sub-series to reduce computational costs while improving local information capture. More recent works, such as TimeMAE (Cheng et al., 2023), enhance this framework by introducing decoupled masked autoencoders, achieving state-of-the-art performance in time series classification tasks. These generative pretraining techniques focus on leveraging reconstruction tasks to learn robust representations for downstream applications.

## 3 METHODOLOGY

### 3.1 PROBLEM DEFINITION

Given an input multivariate time series $X \in \mathbb{R}^{C \times L}$, where $C$ represents the number of channels and $L$ denotes the look-back window length, the objective is to predict future values $Y \in \mathbb{R}^{C \times H}$ over a predicted window $H$. Here, $X = [x_1, \ldots, x_L]$ consists of $L$ input vectors $x_i \in \mathbb{R}^C$, while $Y = [y_{L+1}, \ldots, y_{L+H}]$ represents the predicted values. Initially, we pretrain on the look-back window, and subsequently, both the look-back and prediction windows are employed for the forecasting task.

### 3.2 THE PROPOSED TIMEDART

Our design philosophy centers on integrating two powerful generative approaches: auto-regressive generation and the denoising diffusion model. These two methods complement each other, each leveraging their respective strengths. Auto-Regressive Generation captures the high-level global dependencies within sequence data, while the Denoising Diffusion Model focuses on modeling lower-level local regions. Through their combined efforts, the model learns the deep structures and intrinsic patterns within time series data, ultimately improving prediction accuracy and generalization capability. In the following sections, we will detail the technical aspects of our method.

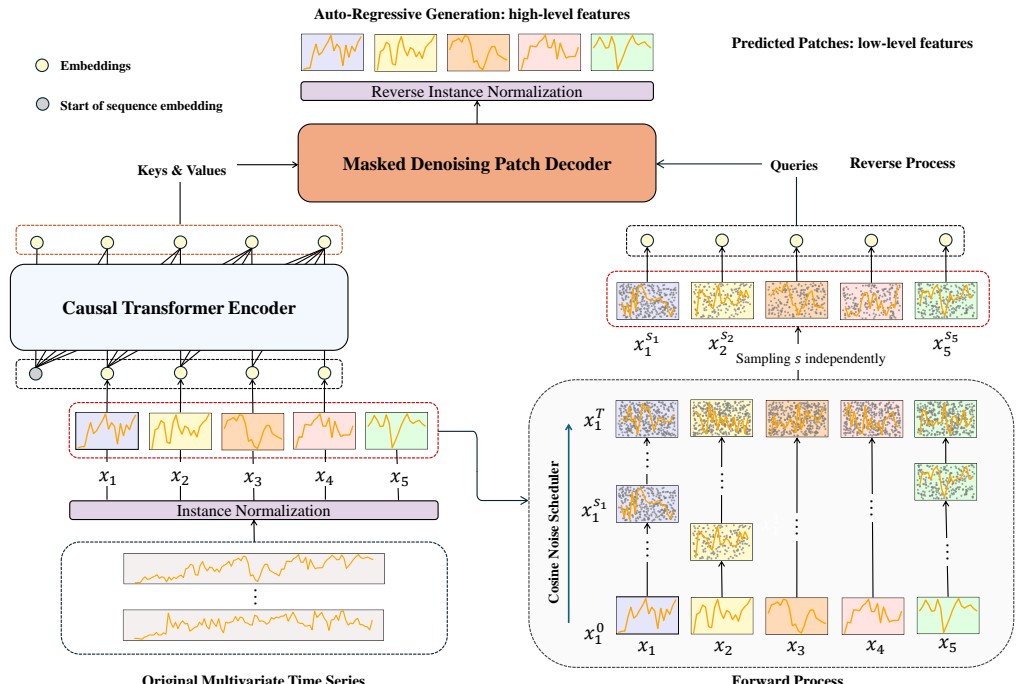

Figure 1: The **TimeDART** architecture captures global dependencies using auto-regressive generation while handling local structures with a denoising diffusion model. The model introduces noise into input patches during the forward diffusion process, generating self-supervised signals. In the reverse process, the original sequence is restored auto-regressively.

### 3.2.1 NORMALIZATION AND PATCHING EMBEDDING

**Instance Normalization.** Before feeding the input multivariate time series data into the representation network, we apply instance normalization to each time series instance $x_{1:L}^{(i)}$, normalizing it to have zero mean and unit standard deviation. After prediction, the original mean and standard deviation are restored to ensure consistency in the final forecast (Kim et al., 2021).

**Channel-Independence.** The input $\boldsymbol{X} = [\boldsymbol{x}_1, \ldots, \boldsymbol{x}_L] \in \mathbb{R}^{C \times L}$ is split to $C$ univariate series $\boldsymbol{x}_{1:L}^{(i)} = [x_1^{(i)}, \ldots, x_L^{(i)}] \in \mathbb{R}^{1 \times L}$ where $i = 1, \ldots, C$. Each of them is fed independently into Transformer encoder. Then the denoising patch decoder will provide results $\boldsymbol{y}_{1:L}^{(i)} = [y_1^{(i)}, \ldots, y_H^{(i)}] \in \mathbb{R}^{1 \times H}$ accordingly. Channel-independence (Zeng et al., 2023; Han et al., 2024) allows universal pre-training across datasets and is common in time series forecasting, enabling different channels to share embedding weights.

**Patching Embedding.** Unlike previous works (Dong et al., 2024; Rasul et al., 2021), we use patches instead of points as the basic modeling unit. This is because patches capture more information and features from local regions, providing richer representations compared to individual points. Additionally, diffusion model operate on these modeling units. Applying noise and denoising to individual points could lead to excessive sensitivity to inherent noise in the dataset, while using patches mitigates this issue by offering a more stable representation. To prevent information leakage and preserve the model's auto-regressive property, we set the patch length $P$ equal to the stride $S$. This ensures that each patch contains only non-overlapping segments of the original sequence, avoiding access to future time steps and maintaining the auto-regressive assumption. For simplicity, we assume $L$, the time series length, is divisible by $P$, resulting in $N = \frac{L}{P}$ patches, which significantly lowers computational complexity and enables the model to process longer sequences.

Each patch (referred to as a clean patch) is then passed through a linear embedding layer, transforming it into a high-dimensional representation. The patch embeddings are expressed as: (we omit the channel index $(i)$ for simplicity):

$$\boldsymbol{z}_{1:N} = \text{Embedding}(\boldsymbol{x}_{1:N}).$$

### 3.2.2 Causal Transformer Encoder

We initialize a vanilla Transformer encoder as the representation network, aligning with existing self-supervised methods. During pre-training, we prepend a learnable start-of-sequence (SOS) embedding to the clean patch representations, while excluding the final one. To further incorporate positional information, we apply sinusoidal positional encoding after the embedding layer. Following this, we use a causal mask $M$ in the self-attention layer, limiting each patch's visibility to itself and prior patches. Let $f(\cdot)$ represent the Transformer encoder's processing of the input sequence with the causal mask, resulting in the final contextualized representations. Consequently, the causal Transformer encoder network can be expressed as follows:

$$\boldsymbol{z}_{1:N}^{in} = \text{Concat}[SOS,\ \boldsymbol{z}_{1:N-1}] + \text{PE}_{1:N},$$

$$f(\boldsymbol{z}_{1:N}^{in}) = \text{Encoder}(\boldsymbol{z}_{1:N}^{in},\ M).$$

### 3.2.3 Prediction with Diffusion Generation

Different from previous self-supervised learning (SSL) approaches, our work innovatively incorporates the diffusion model into self-supervised prediction. The diffusion model consists of two key steps: the forward process and the reverse denoising (Shen et al., 2024; Fan et al., 2024; Li et al.; Yuan & Qiao, 2024). The forward process gradually adds noise to the data, while the reverse process reconstructs the original data by removing the noise. Below, we detail the techniques of this approach.

**Forward Process.** For each patch $x_j \in \boldsymbol{x}_{1:N}$, the forward process $q(x_j^s|x_j^{s-1}) = \mathcal{N}(x_j^s; \sqrt{\alpha(s)}x_j^{s-1}, (1-\alpha(s))I)$ gradually adds noise to the patch, where $\alpha(s)$ is the noise scheduler. Let $\gamma(s)$ be the cumulative product of $\alpha$ over time steps, where $\gamma(s) = \prod_{s' \leq s} \alpha(s')$, the forward process can be rewrite given the original clean patch $x_j^0$:

$$q(x_j^s|x_j^0) = \mathcal{N}(x_j^s; \sqrt{\gamma(s)}x_j^0, (1-\gamma(s))I).$$

As shown in Figure 1, we independently add noise to each patch at time step $s$, enabling the model to learn varying denoising scales across the sequence. This prevents oversimplification of the task, ensuring robust pre-training. The resulting sequence of noisy patches is represented as:

$$\hat{\boldsymbol{x}}_{1:N} = [x_1^{s_1}, \ldots, x_N^{s_N}],$$

In DDPM (Ho et al., 2020), the noise scheduler $\alpha(s)$ typically decreases linearly as $s$ increases. Instead, we use a cosine scheduling approach, where $\alpha(s) \propto \cos\left(\frac{s}{T}\pi\right)$. This smoother transition emphasizes the early and later stages of diffusion, improving model stability and better capturing the data distribution.

The noise-added and clean patches share the same embedding layer and weights. Both also use sinusoidal positional encoding. The deep representation of the noise-added patches is as follows:

$$\hat{\boldsymbol{z}}_{1:N}^{in} = \text{Embedding}(\hat{\boldsymbol{x}}_{1:N}) + \text{PE}_{1:N}.$$

**Reverse Process.** The reverse process is handled by the denoising patch decoder, which is a Transformer Decoder block. It takes the Transformer encoder output as keys and values, while the noise-added patch embeddings act as queries.

A mask is applied to the decoder to ensure that the $j$-th input in the noise-added sequence can only attend to the $j$-th output from the Transformer encoder. The encoder's output at position $j$, informed by the causal mask and start-of-sequence (SOS) embedding, aggregates information from

clean patches at positions $1$ to $j-1$, enabling auto-regressive generation. Finally, deep representations are mapped back to the original space via flattening and linear projection. Although the linear layer concatenates the generated sequence and projects it into the input space, this does not imply that the auto-regressive mechanism is irrelevant. We will demonstrate the effectiveness of the auto-regressive mechanism through subsequent experiments by removing the Causal Mask in the Transformer encoder and the mask in the denoising patch decoder in Section D.

Let $g(\cdot)$ denote the processing of the two inputs by the denoising patch decoder. The reverse process is then expressed as follows::

$$z_j^{out} = g(\hat{z}_j^{in}, \ f(\boldsymbol{z}_{1:j-1}^{in})), \quad 1 \le j \le N.$$

### 3.3 Self-supervised Generative Optimization

Instead of using a masked optimization approach, we adopt an auto-regressive generative scheme for several reasons. First, generative models are better suited for prediction tasks. For example, GPT (Radford, 2018) is favored over BERT (Devlin, 2018) in conversational models due to its superior performance in sequential prediction, making it a better fit for generating future outcomes. Second, while masked modeling captures bidirectional context, it introduces inconsistencies between pre-training and downstream tasks. Masked token embeddings exist only in pre-training, causing a mismatch during fine-tuning. Additionally, pre-training exposes the model to partial data (with masked tokens), whereas downstream tasks use full sequences, further exacerbating this discrepancy.

We also replace the conventional MSE loss with a denoising diffusion model and its diffusion loss. Diffusion loss helps the model capture multimodal distributions, better suited for the complexity of time series data. In contrast, MSE assumes predicted values center around a single mean, often resulting in overly smooth predictions that fail to capture the multimodal patterns in time series data.

Our self-supervised optimization objective minimizes the diffusion loss, equivalent to the Evidence Lower Bound (ELBO). The final loss is:

$$\mathcal{L}_{diff} = \mathcal{L}_{ELBO} = \sum_{j=1}^{N} \mathbb{E}_{\epsilon, q(x_j^0)} \left[ ||x_j^0 - g(\hat{z}_j^{in}, \ f(\boldsymbol{z}_{1:j-1}^{in}))||^2 \right].$$

The detailed derivation process can be found in Appendix B.

#### 3.3.1 Downstream Transfering

After pre-training on the look-back window, fine-tuning is performed on both the look-back and predicted windows by re-initializing a new prediction head for the downstream task and removing the denoising patch decoder. During fine-tuning, the model is optimized for one-step prediction using MSE loss. This approach maintains structural consistency between pre-training and downstream tasks, while keeping their objectives distinct.

## 4 Experiments

### 4.1 Experimental Setup

**Datasets.** To evaluate TimeDART, we conduct experiments on 8 popular datasets, including 4 ETT datasets (ETTh1, ETTh2, ETTm1, ETTm2), Weather, Exchange, Electricity, and Traffic. The statistics of these datasets are summarized in Table 1. Following standard protocol, we split each dataset into training, validation, and testing sets in chronological order. The split ratio is $6:2:2$ for the ETT datasets and $7:1:2$ for the others.

**Baselines and Experimental Settings.** Since we adopted the channel-independence setting, we can perform general pre-training across all eight datasets. Therefore, we conducted two experimental settings: in-domain and cross-domain. In the in-domain setting, both pre-training and fine-tuning

Table 1: The Statistics of Each Dataset.

| Dataset | Variables | Frequency | Length | Scope |
|---------|-----------|-----------|--------|-------|
| ETTh1/ETTh2 | 7 | 1 Hour | 17420 | Energy |
| ETTm1/ETTm2 | 7 | 15 Minutes | 69680 | Energy |
| Electricity | 321 | 1 Hour | 26304 | Energy |
| Traffic | 862 | 1 Hour | 17544 | Transportation |
| Weather | 21 | 10 Minutes | 52696 | Weather |
| Exchange | 8 | 1 Day | 7588 | Finance |

were performed on the same dataset, whereas in the cross-domain setting, we pre-trained on five datasets (ETTh1, ETTh2, ETTm1, ETTm2, Electricity) from the Energy domain and fine-tuned on a specific dataset.

We compared our approach against several state-of-the-art baseline methods. In the in-domain setting, we selected six competitive methods, along with results from a randomly initialized model for comparison. Among them, **SimMTM** (Dong et al., 2024) proposes recovering masked time points by weighted aggregation of multiple neighbors outside the manifold, while also utilizing contrastive learning to optimize the self-supervised process. **PatchTST** (Nie et al., 2022) in its self-supervised version leverages subseries-level patches and channel-independence to retain local semantics, reduce computation, and enhance long-term forecasting accuracy. Additionally, **TimeMAE** (Cheng et al., 2023) utilizes decoupled masked autoencoders to learn robust representations for regression. **CoST** (Woo et al., 2022) is a time series forecasting framework that uses contrastive learning to disentangle seasonal and trend representations. Furthermore, we compared against supervised methods, such as the supervised version of the Transformer-based **PatchTST** (Nie et al., 2022) and the linear-based **DLinear** (Zeng et al., 2023) model, to further demonstrate the effectiveness of TimeDART.

In the cross-domain setting, we perform mixed pre-training on five datasets [ETTh1, ETTh2, ETTm1, ETTm2, Electricity] from the Energy domain, followed by fine-tuning on a specific dataset from these five. The cross-domain baseline includes the results from a randomly initialized model and the performance of TimeDART in the in-domain setting.

**Fair Experiment.** To ensure experimental fairness, we used a unified encoder for all representation networks in the in-domain setting, except for DLinear. Specifically, we adopted a vanilla Transformer encoder with a channel-independent configuration, while DLinear retained its native linear encoder settings. All implementations are based on their official repositories.

Similarly, to ensure fairness, we set the lookback window length $L = 336$ and the predicted window $H \in \{96, 192, 336, 720\}$, following the standard protocol. To highlight the differences introduced by pre-training, we also include a random init setting, where the representation network is randomly initialized and then fine-tuned on the same downstream tasks without any pre-training. This setup clearly demonstrates the significant improvements brought by pre-training.

## 4.2 MAIN RESULT

The experimental results for the in-domain setting are shown in Table 2, while the results for the cross-domain setting are shown in Table 3.

After downstream fine-tuning, TimeDART outperforms its competing baselines in most experimental settings, achieving the best results in approximately **67%** of the 64 evaluation metrics. Specifically, TimeDART surpasses the best baselines across all metrics in the ETTh2 and ETTm2 datasets, consistently outperforming both self-supervised and supervised methods. TimeDART also demonstrates significant advantages due to pre-training, as seen in its superior performance compared to non-pre-trained baselines across all datasets and prediction horizons. Although it may not always achieve the top result, TimeDART consistently ranks as either the best or second-best method in nearly all settings, with only four exceptions. The method shows relatively weaker performance on the Exchange dataset, primarily due to the uneven distribution between the look-back and predicted windows, which limits its ability to fully exploit its strengths in balancing upstream and downstream input data. Furthermore, the marked differences in data trends between the validation and test sets in this dataset lead to overfitting, necessitating more effective generalization strategies for such cases.

To clearly demonstrate the effectiveness of our method, the visualized prediction results will be presented in Section F.

Table 2: Multivariate time series forecasting results comparing TimeDART with both SOTA self-supervised approaches and supervised approaches. The best results are in **bold** and the second best are underlined. "#1 Counts" represents the number of times the method achieves the best results.

| Methods | | Ours | | | | Self-supervised | | | | | | | | Supervised | | | |
|---|---|---|---|---|---|---|---|---|---|---|---|---|---|---|---|---|---|
| | | **TimeDART** | | Random Init. | | SimMTM | | PatchTST | | TimeMAE | | CoST | | PatchTST | | DLinear | |
| Metric | | MSE | MAE | MSE | MAE | MSE | MAE | MSE | MAE | MSE | MAE | MSE | MAE | MSE | MAE | MSE | MAE |
| ETTh1 | 96 | **0.370** | **0.395** | 0.383 | 0.405 | 0.379 | 0.407 | 0.384 | 0.401 | 0.387 | 0.411 | 0.422 | 0.436 | 0.382 | 0.403 | _0.375_ | _0.396_ |
| | 192 | **0.402** | **0.419** | 0.439 | 0.439 | _0.412_ | 0.424 | 0.427 | 0.431 | 0.420 | 0.431 | 0.520 | 0.487 | 0.416 | _0.423_ | 0.428 | 0.437 |
| | 336 | _0.426_ | **0.427** | 0.467 | 0.457 | **0.421** | _0.431_ | 0.461 | 0.450 | 0.453 | 0.453 | 0.472 | 0.462 | 0.441 | 0.440 | 0.448 | 0.449 |
| | 720 | _0.446_ | _0.462_ | 0.468 | 0.475 | **0.424** | **0.449** | 0.460 | 0.465 | 0.476 | 0.485 | 0.525 | 0.501 | 0.470 | 0.475 | 0.505 | 0.514 |
| ETTh2 | 96 | **0.283** | **0.340** | 0.294 | 0.348 | 0.293 | 0.347 | 0.297 | 0.354 | 0.325 | 0.378 | 0.321 | 0.374 | _0.286_ | _0.342_ | 0.296 | 0.360 |
| | 192 | **0.343** | **0.381** | 0.357 | 0.390 | _0.355_ | _0.386_ | 0.388 | 0.406 | 0.394 | 0.423 | 0.380 | 0.403 | 0.357 | 0.389 | 0.391 | 0.423 |
| | 336 | **0.364** | **0.399** | 0.375 | 0.408 | _0.370_ | _0.401_ | 0.392 | 0.413 | 0.424 | 0.447 | 0.430 | 0.451 | 0.377 | 0.409 | 0.445 | 0.460 |
| | 720 | **0.390** | **0.425** | 0.407 | 0.439 | _0.395_ | _0.427_ | 0.413 | 0.442 | 0.464 | 0.476 | 0.466 | 0.480 | 0.406 | 0.440 | 0.700 | 0.592 |
| ETTm1 | 96 | **0.286** | **0.342** | 0.301 | 0.354 | _0.288_ | 0.348 | 0.289 | 0.344 | 0.289 | 0.344 | 0.291 | _0.343_ | 0.298 | 0.345 | 0.303 | 0.346 |
| | 192 | **0.326** | **0.367** | 0.333 | 0.372 | _0.327_ | 0.373 | **0.326** | 0.372 | 0.33 | 0.371 | 0.330 | 0.370 | 0.339 | 0.374 | 0.338 | _0.368_ |
| | 336 | _0.357_ | _0.388_ | 0.360 | 0.389 | 0.363 | 0.395 | **0.353** | **0.387** | 0.366 | 0.393 | 0.382 | 0.401 | 0.381 | 0.401 | 0.373 | 0.393 |
| | 720 | _0.407_ | **0.417** | 0.408 | _0.418_ | 0.412 | 0.424 | **0.399** | _0.418_ | 0.416 | 0.424 | 0.422 | 0.425 | 0.428 | 0.431 | 0.428 | 0.423 |
| ETTm2 | 96 | **0.165** | **0.256** | 0.174 | 0.263 | 0.172 | 0.261 | 0.171 | _0.257_ | 0.174 | 0.263 | 0.242 | 0.333 | 0.174 | 0.261 | _0.170_ | 0.264 |
| | 192 | **0.221** | **0.294** | 0.240 | 0.307 | _0.223_ | _0.300_ | 0.236 | 0.304 | 0.233 | 0.303 | 0.283 | 0.345 | 0.238 | 0.307 | 0.233 | 0.311 |
| | 336 | **0.279** | **0.330** | 0.284 | 0.334 | _0.282_ | _0.331_ | 0.291 | 0.344 | 0.291 | 0.340 | 0.303 | 0.349 | 0.293 | 0.346 | 0.298 | 0.358 |
| | 720 | **0.364** | **0.385** | 0.377 | 0.389 | 0.374 | _0.388_ | 0.388 | 0.404 | 0.380 | 0.396 | 0.431 | 0.431 | _0.373_ | 0.401 | 0.423 | 0.437 |
| Electricity | 96 | **0.132** | _0.225_ | 0.147 | 0.252 | _0.133_ | 0.223 | **0.132** | _0.225_ | 0.165 | 0.285 | 0.197 | 0.277 | 0.138 | **0.233** | 0.141 | 0.238 |
| | 192 | 0.150 | _0.241_ | 0.163 | 0.265 | **0.147** | **0.237** | _0.148_ | _0.241_ | 0.181 | 0.297 | 0.197 | 0.279 | 0.153 | 0.247 | 0.154 | 0.251 |
| | 336 | **0.166** | **0.258** | 0.179 | 0.280 | **0.166** | 0.265 | _0.167_ | _0.260_ | 0.199 | 0.312 | 0.211 | 0.295 | 0.170 | 0.263 | 0.170 | 0.269 |
| | 720 | **0.203** | **0.290** | 0.218 | 0.312 | **0.203** | 0.297 | _0.205_ | _0.292_ | 0.238 | 0.341 | 0.255 | 0.330 | 0.206 | 0.295 | _0.205_ | 0.302 |
| Traffic | 96 | **0.357** | **0.247** | 0.386 | 0.267 | _0.368_ | 0.262 | 0.382 | 0.262 | 0.382 | _0.261_ | 0.378 | 0.365 | 0.395 | 0.272 | 0.411 | 0.284 |
| | 192 | 0.376 | _0.256_ | 0.398 | 0.267 | _0.373_ | **0.251** | 0.385 | 0.261 | 0.399 | 0.267 | **0.371** | 0.352 | 0.411 | 0.278 | 0.423 | 0.289 |
| | 336 | **0.389** | _0.262_ | 0.410 | 0.274 | _0.395_ | **0.254** | 0.409 | 0.275 | 0.411 | 0.274 | 0.467 | 0.354 | 0.424 | 0.284 | 0.437 | 0.297 |
| | 720 | **0.429** | **0.286** | 0.446 | 0.299 | _0.432_ | _0.290_ | 0.438 | 0.291 | 0.446 | 0.298 | 0.525 | 0.378 | 0.453 | 0.300 | 0.467 | 0.316 |
| Weather | 96 | 0.149 | 0.199 | 0.155 | 0.206 | 0.158 | 0.211 | _0.148_ | **0.196** | 0.150 | 0.203 | 0.216 | 0.280 | **0.147** | _0.197_ | 0.176 | 0.236 |
| | 192 | _0.193_ | **0.240** | 0.198 | 0.246 | 0.199 | 0.249 | _0.193_ | **0.240** | **0.191** | _0.241_ | 0.303 | 0.335 | **0.191** | **0.240** | 0.217 | 0.275 |
| | 336 | _0.244_ | _0.280_ | 0.250 | 0.286 | 0.246 | 0.286 | _0.244_ | **0.279** | **0.243** | 0.282 | 0.351 | 0.358 | _0.244_ | 0.282 | 0.264 | 0.315 |
| | 720 | **0.317** | **0.331** | 0.319 | 0.335 | **0.317** | 0.337 | 0.321 | _0.334_ | _0.318_ | _0.334_ | 0.425 | 0.343 | 0.320 | _0.334_ | 0.325 | 0.364 |
| Exchange | 96 | **0.086** | _0.211_ | 0.102 | 0.229 | 0.100 | 0.226 | 0.088 | **0.207** | 0.098 | 0.226 | 0.102 | 0.229 | 0.094 | 0.213 | _0.087_ | 0.217 |
| | 192 | _0.175_ | _0.302_ | 0.224 | 0.343 | 0.210 | 0.332 | 0.186 | 0.308 | 0.219 | 0.340 | 0.212 | 0.334 | 0.191 | 0.311 | **0.164** | **0.298** |
| | 336 | 0.344 | _0.431_ | 0.384 | 0.453 | 0.389 | 0.460 | 0.374 | 0.446 | 0.400 | 0.466 | 0.384 | 0.452 | _0.343_ | **0.427** | **0.333** | 0.437 |
| | 720 | **0.829** | **0.675** | 1.051 | 0.774 | 1.104 | 0.800 | _0.857_ | _0.692_ | 0.989 | 0.751 | 1.124 | 0.805 | 0.888 | 0.706 | 0.988 | 0.749 |
| #1 Counts | | 43 | | 0 | | 10 | | 9 | | 2 | | 1 | | 5 | | 3 | |

Table 3: Multivariate time series forecasting results comparing TimeDART, pretrained across five datasets and fine-tuned on specific ones. All results are averaged from 4 different predicted window of {96, 192, 336, 720}. The best results are in **bold**. See Appendix C for full results.

| Methods | **TimeDART (CD)** | | Random Init.(CD) | | TimeDART (ID) | | Random Init. (ID) | |
|---|---|---|---|---|---|---|---|---|
| Metric | MSE | MAE | MSE | MAE | MSE | MAE | MSE | MAE |
| ETTh1 | **0.409** | 0.429 | 0.430 | 0.442 | 0.411 | **0.426** | 0.439 | 0.444 |
| ETTh2 | **0.343** | **0.385** | 0.363 | 0.405 | 0.345 | 0.386 | 0.358 | 0.396 |
| ETTm1 | 0.348 | 0.381 | 0.355 | 0.386 | **0.344** | **0.379** | 0.351 | 0.383 |
| ETTm2 | **0.256** | **0.315** | 0.269 | 0.323 | 0.257 | 0.316 | 0.269 | 0.323 |
| Electricity | **0.162** | **0.254** | 0.166 | 0.259 | 0.163 | **0.254** | 0.177 | 0.277 |

As shown in Table 3, the overall effectiveness of TimeDART in cross-domain scenarios is evident. TimeDART consistently outperforms the random initialization baseline, demonstrating its strong ability to generalize across diverse time series datasets. The use of cross-domain pre-training leads

to improved forecasting accuracy by learning robust representations from multiple datasets. For instance, on the ETTh2 dataset, TimeDART's cross-domain pre-training significantly surpasses in-domain training, illustrating the benefits of leveraging varied temporal patterns and dependencies from different datasets. In contrast, the ETTm2 dataset presents a more challenging scenario, where the distinct characteristics of the data make cross-domain pre-training less effective. However, even in this case, the performance difference between cross-domain and in-domain training remains minimal, showing that TimeDART maintains competitive performance even in more difficult settings. Overall, the experiments demonstrate TimeDART's ability to enhance generalization across datasets while handling varying distributional characteristics.

## 4.3 ABLATION STUDY

We investigated the effectiveness of two key modules: the auto-regressive generation and the denoising diffusion model. Four experimental settings were considered: the original model, named TimeDART, the model with the auto-regressive generation removed, named $w/o$ $AR$, the model without the denoising diffusion process, named $w/o$ $diff$, and the model with both modules removed, named $w/o$ $AR$-$diff$. Specifically, in the auto-regressive removal experiment, we eliminated both the causal mask in the Transformer encoder and the mask in the denoising patch decoder. In the denoising patch decoder removal experiment, we bypassed the noise addition and denoising process, allowing the output of the representation network to directly pass into the linear projection layer.

Table 4: The ablation study. All results are averaged from 4 different predicted window of $\{96, 192, 336, 720\}$. The best results are in **bold**. See Appendix D for full results.

| Metric | TimeDART MSE | TimeDART MAE | $W/o$ AR MSE | $W/o$ AR MAE | $W/o$ Diff MSE | $W/o$ Diff MAE | $W/o$ AR-Diff MSE | $W/o$ AR-Diff MAE |
|---|---|---|---|---|---|---|---|---|
| ETTh2 | **0.346** | **0.387** | 0.365 | 0.399 | 0.352 | 0.391 | 0.364 | 0.398 |
| ETTm2 | **0.257** | **0.316** | 0.281 | 0.338 | 0.265 | 0.322 | 0.285 | 0.346 |
| Electricity | **0.163** | **0.254** | 0.193 | 0.304 | 0.164 | 0.255 | 0.190 | 0.299 |

Table 4 demonstrate that both the auto-regressive generation and the denoising diffusion model play crucial roles in the effectiveness of this approach. Notably, removing the auto-regressive mechanism leads to performance that is even worse than random initialization, further confirming our claim in the method section that the final linear projection layer does not diminish the impact of the auto-regressive mechanism.

## 4.4 HYPERPARAMETER SENSITIVITY ANALYSIS

In our hyperparameter sensitivity experiments, we first investigate two key parameters: the total number of diffusion steps $T \in \{750, 1000, 1250\}$ and the noise scheduler $\alpha(s)$, comparing cosine and linear schedules. The number of diffusion steps reflects the pre-training difficulty, with higher $T$ values making it harder to recover clean patches. The noise scheduler controls the smoothness of noise addition, with the cosine scheduler providing smoother transitions than the linear one. These experiments are conducted on both the ETTh2 and ETTm2 datasets, as shown in Table 5. For brevity, we report the results as the mean across four prediction lengths.

Table 5: Hyperparameter sensitivity snalysis of total noise steps and noise schedulers. All results are averaged from 4 different predicted window of $\{96, 192, 336, 720\}$. The best results are in **bold**. See Appendix E.1 for full results.

| | ETTh2 | | | | | | ETTm2 | | | | | |
|---|---|---|---|---|---|---|---|---|---|---|---|---|
| | (a) Total Noise Steps | | | (b) Noise Scheduler | | | (a) Total Noise Steps | | | (b) Noise Scheduler | | |
| Value | MSE | MAE | Type | MSE | MAE | Value | MSE | MAE | Type | MSE | MAE |
| 750 | 0.349 | 0.393 | Cos. | **0.345** | **0.386** | 750 | 0.263 | 0.322 | Cos. | **0.257** | **0.316** |
| 1000 | **0.345** | **0.386** | Lin. | 0.358 | 0.396 | 1000 | **0.257** | **0.316** | Lin. | 0.369 | 0.323 |
| 1250 | 0.347 | 0.391 | | | | 1250 | 0.263 | 0.321 | | | |

As shown in Table 5, the total number of noise steps does not significantly impact the difficulty of pre-training. However, calculations indicate that models pre-trained with different noise steps still outperform those with random initialization. Notably, the cosine noise scheduler performs substantially better than the linear scheduler. In some cases, using the linear scheduler even leads to results worse than those from random initialization. This highlights the critical importance of the noise scheduler, as insufficiently smooth noise addition can result in significantly poorer outcomes.

We then evaluate the impact of the number of layers in the denoising patch decoder across the ETTh2, ETTm2, and Electricity datasets. The number of layers, selected from $[0, 1, 2, 3]$, reflects the relative size of the denoising network compared to the representation network, which is fixed at 2 layers for all datasets. A decoder with 0 layers represents an ablation case where the denoising patch decoder is removed in Section 4.3. As can be observed in Figure 2, allocating too many layers to the denoising patch decoder can lead to under-training of the representation network, as the majority of the model's parameters are concentrated in the denoising component.

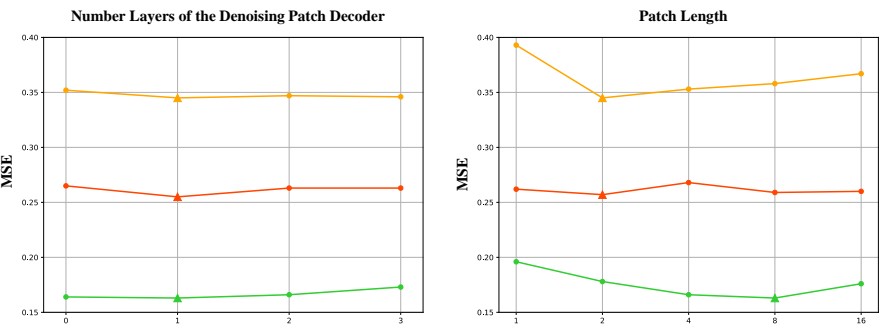

Figure 2: Hyperparameter analysis of number layers of denoising patch decoder and patch length in TimeDART. All results are averaged from 4 different predicted window of $\{96, 192, 336, 720\}$. The triangle symbol represents the best prediction. See Appendix E.2 and E.3 for full results.

Finally we examine the effect of patch length, selected from $[1, 2, 4, 8, 16]$, which controls the amount of local segment information each patch carries. Patch length determines the scale of intra-patch information, and its optimal value depends on the redundancy between neighboring data points within each dataset. For example, in datasets like Electricity, which exhibit higher redundancy between consecutive data points, larger patch lengths may be more effective for modeling. Conversely, for datasets with less redundancy between adjacent data points, shorter patch lengths may be preferred to capture finer-grained temporal dynamics. Figure 2 indicates that different datasets require different levels of intra-patch analysis, reinforcing the need for adaptive patch length selection based on dataset characteristics.

## 5 CONCLUSION

In this paper, we proposed TimeDART, a novel generative self-supervised method for time series forecasting that effectively captures both global sequence dependencies and local detail features. By treating time series patches as basic modeling units, TimeDART employs a self-attention-based Transformer encoder to model the sequence dependencies between patches. Simultaneously, it incorporates diffusion and denoising mechanisms to capture the locality features within each patch. Notably, our design of a cross-attention-based flexible denoising network allows for adjustable optimization difficulty in the self-supervised task, enhancing the model's learning effectiveness. Extensive experiments demonstrate that TimeDART achieves state-of-the-art fine-tuning performance compared to existing advanced time series pre-training methods in forecasting tasks.

## 6 REPRODUCIBILITY STATEMENT

In the main text, we have clearly described the architecture of the TimeDART with detailed equations. All implementation details are thoroughly provided in the Appendix, including comprehensive descriptions of the datasets, experimental settings, evaluation metrics, and hyperparameters used in our experiments. Additionally, the training procedures and data preprocessing steps are documented for transparency. The source code, along with all necessary scripts for replicating the experiments, has already been made publicly available and can be accessed through the provided anonymous link [2].

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

# A  IMPLEMENTATION DETAILS

## A.1  DATASET DESCRIPTIONS

We conducted extensive experiments on eight real-world datasets to evaluate the effectiveness of the proposed TimeDART method under both in-domain and cross-domain settings. These datasets cover a variety of application scenarios, including power systems, transportation networks, and weather forecasting. For detailed descriptions of the datasets and their respective divisions, please refer to Table 6.

Table 6: Dataset descriptions. *Samples* are organized in (Train/Validation/Test).

| Dataset | Variables | Predicted Window | Samples | Scope | Frequency |
|---------|-----------|------------------|---------|-------|-----------|
| ETTh1,ETTh2 | 7 | {96,192,336,720} | 8209/2785/2785 | Energy | 1 Hour |
| ETTm1,ETTm2 | 7 | {96,192,336,720} | 34129/11425/11425 | Energy | 15 Mins |
| Electricity | 321 | {96,192,336,720} | 17981/2537/5165 | Energy | 1 Hour |
| Traffic | 862 | {96,192,336,720} | 11849/1661/3413 | Transportation | 1 Hour |
| Weather | 21 | {96,192,336,720} | 36456/5175/10444 | Weather | 10 Mins |
| Exchange | 8 | {96,192,336,720} | 4880/665/1422 | Finance | 1 Day |

**ETT (4 subsets)** (Zhou et al., 2021): This dataset comprises time series data of oil temperature and power load collected from electricity transformers spanning July 2016 to July 2018. It is divided into four subsets, each with different recording intervals: ETTh1 and ETTh2 have hourly recordings, while ETTm1 and ETTm2 are recorded every 15 minutes.

**Electricity** (UCI): This dataset captures the electricity consumption of 321 clients on an hourly basis from 2012 to 2014, with measurements taken every 15 minutes (in kW). Time stamps follow Portuguese time. Each day includes 96 measurements (24×4), and during time changes in March (where one hour is skipped), the values between 1:00 am and 2:00 am are set to zero. Conversely, in October (with an extra hour), consumption between 1:00 am and 2:00 am represents the aggregated values of two hours.

**Traffic** (PeMS): Road occupancy rates, measured hourly, were collected from 862 sensors located along the San Francisco Bay area freeways. The data spans from January 2015 to December 2016.

**Weather** (Wetterstation): This dataset contains meteorological time series featuring 21 indicators. The data was collected every 10 minutes in 2020 by the Weather Station at the Max Planck Biogeo-chemistry Institute.

**Exchange** (Guokun Lai): This dataset collects the daily exchange rates of eight countries—Australia, the UK, Canada, Switzerland, China, Japan, New Zealand, and Singapore—from 1990 to 2016.

## A.2  IMPLEMENTATION DETAILS

All experiments were implemented using PyTorch (Paszke et al., 2017) and executed on a single NVIDIA RTX 4090 16GB GPU. For both pre-training and fine-tuning, we employed the ADAM optimizer (Kingma & Ba, 2017), with initial learning rates selected from $\{10^{-3}, 5 \times 10^{-4}, 10^{-4}\}$, and optimized the model using L2 loss. For in-domain pre-training, we set the batch size to 16 for all datasets except Traffic, where it is reduced to 8 due to memory and time limitations. The representation network consists of 2 layers across most datasets, while for Traffic, it has 3 layers. The pre-training process spans 50 epochs, except for Traffic, where it is limited to 30 epochs. In downstream tasks, the settings remain largely the same, except that fine-tuning is performed for 10 epochs. The sequence representation dimension is chosen from $\{8, 16, 32, 64, 128\}$. For cross-domain experiments, the settings mirror those of the Electricity dataset.

# B   OPTIMIZATION OBJECTIVE DERIVATION DETAILS

The self-supervised optimization objective we employ follows the classical form of diffusion loss, which is designed to maximize the marginal likelihood of the data $p(\boldsymbol{x}_0)$. In this context, we assume that $p$ represents the reverse denoising process, where the model learns to reconstruct the original data $\boldsymbol{x}_0$ from its noisy versions. This denoising process is modeled as a gradual reverse transformation of the corrupted data, recovering the underlying clean distribution. The ideal loss function for this process can be formally expressed as:

$$\mathcal{L}_{ideal} = \sum_{j=1}^{N} H(p_\theta(x_j^0),\ q(x_j^0)) = \sum_{j=1}^{N} \mathbb{E}_{q(x_j^0)}[-\log p_\theta(x_j^0)]$$

However, since directly optimizing the exact marginal likelihood is intractable, we instead minimize the Evidence Lower Bound (ELBO), given by:

$$\mathcal{L}_{ideal} \leq \mathcal{L}_{ELBO} = \sum_{j=1}^{N} \mathbb{E}_{q(x_j^{0:T})} \left[ -\log \frac{q(x_j^{1:T}|x_j^0)}{p_\theta(x_j^{0:T})} \right]$$

Following a series of derivations (Luo, 2022), the final loss function is:

$$\mathcal{L}_{diff} = \mathcal{L}_{ELBO} = \sum_{j=1}^{N} \mathbb{E}_{\epsilon, q(x_j^0)} \left[ ||x_j^0 - g(\hat{z}_j^{in},\ f(\boldsymbol{z}_{1:j-1}^{in}))||^2 \right],$$

# C   CROSS DOMAIN FULL RESULT

The results in Table 7 demonstrate that TimeDART consistently outperforms random initialization across all datasets and prediction lengths. For ETTh2, TimeDART (CD) achieves the lowest MSE of 0.280 at the 96-step window and maintains superior performance over longer horizons, consistently surpassing both random initialization and in-domain training. At the 192-step window, it records an MSE of 0.342 and MAE of 0.380, compared to random initialization's MSE of 0.358 and MAE of 0.398, further emphasizing the benefits of cross-domain pre-training. For ETTm2, cross-domain pre-training provides a distinct advantage, particularly at the 336-step horizon, where TimeDART (CD) outperforms TimeDART (ID) by 0.05 in MSE. This highlights the model's robustness in longer forecasting windows. While the cross-domain approach generally surpasses in-domain training, certain datasets, such as ETTm1, present challenges due to distributional differences. However, the performance gap remains small.

# D   ABLATION STUDY

The results in Table 8 underscore the critical roles of both the auto-regressive generation and denoising diffusion components in TimeDART. Removing the auto-regressive mechanism (*w/o AR*) leads to a significant performance decline, particularly in ETTm2. At the 96-step horizon, MSE increases from 0.165 to 0.184, and at 336 steps, it rises from 0.279 to 0.307. This illustrates the crucial role of the auto-regressive mechanism in enhancing the model's forecasting ability, especially across various time horizons. Similarly, eliminating the denoising diffusion module (*w/o Diff*) results in noticeable performance degradation, as observed in ETTh2. At the 96-step horizon, MSE increases from 0.283 to 0.288, and at the 336-step horizon, it rises from 0.365 to 0.372. These findings highlight the essential contribution of the denoising diffusion process to improving the model's learning and overall performance.

When both components are removed (*w/o AR-Diff*), the model's performance deteriorates significantly across all datasets. For instance, in Electricity, at the 336-step horizon, MSE jumps from 0.166 to 0.199, clearly showing the combined importance of both modules for achieving optimal performance.

Table 7: Full result of Multivariate time series forecasting results comparing TimeDART, pretrained across five datasets and fine-tuned on specific ones. All results are conducted on 4 different predicted window of $\{96, 192, 336, 720\}$. The best results are in **bold**.

| Methods | | **TimeDART (CD)** | | Random Init.(CD) | | TimeDART (ID) | | Random Init. (ID) | |
| Metric | | MSE | MAE | MSE | MAE | MSE | MAE | MSE | MAE |
|---|---|---|---|---|---|---|---|---|---|
| | 96 | **0.365** | **0.394** | 0.378 | 0.402 | 0.370 | 0.395 | 0.383 | 0.405 |
| | 192 | **0.399** | **0.418** | 0.421 | 0.428 | 0.402 | 0.419 | 0.439 | 0.439 |
| ETTh1 | 336 | 0.430 | 0.438 | 0.434 | 0.444 | **0.426** | **0.427** | 0.467 | 0.457 |
| | 720 | **0.442** | 0.467 | 0.488 | 0.493 | 0.446 | 0.462 | 0.468 | 0.475 |
| | Avg. | **0.409** | 0.429 | 0.430 | 0.442 | 0.411 | **0.426** | 0.439 | 0.444 |
| | 96 | **0.280** | **0.339** | 0.294 | 0.353 | 0.283 | **0.340** | 0.294 | 0.348 |
| | 192 | **0.342** | **0.380** | 0.358 | 0.398 | 0.343 | 0.381 | 0.357 | 0.390 |
| ETTh2 | 336 | **0.362** | **0.398** | 0.386 | 0.423 | 0.364 | 0.399 | 0.375 | 0.408 |
| | 720 | **0.388** | **0.424** | 0.413 | 0.444 | 0.390 | 0.425 | 0.407 | 0.439 |
| | Avg. | **0.343** | **0.385** | 0.363 | 0.405 | 0.345 | 0.386 | 0.358 | 0.396 |
| | 96 | 0.287 | **0.342** | 0.292 | 0.346 | **0.286** | **0.342** | 0.301 | 0.354 |
| | 192 | **0.325** | **0.366** | 0.335 | 0.371 | 0.326 | 0.367 | 0.333 | 0.372 |
| ETTm1 | 336 | 0.367 | 0.395 | 0.370 | 0.395 | **0.357** | **0.388** | 0.360 | 0.389 |
| | 720 | 0.411 | 0.420 | 0.422 | 0.430 | **0.407** | **0.417** | 0.408 | 0.418 |
| | Avg. | 0.348 | 0.381 | 0.355 | 0.386 | **0.344** | **0.379** | 0.351 | 0.383 |
| | 96 | **0.165** | **0.255** | 0.174 | 0.263 | **0.165** | 0.256 | 0.174 | 0.263 |
| | 192 | 0.222 | **0.293** | 0.240 | 0.307 | **0.221** | 0.294 | 0.240 | 0.307 |
| ETTm2 | 336 | **0.274** | **0.328** | 0.284 | 0.334 | 0.279 | 0.330 | 0.284 | 0.334 |
| | 720 | **0.361** | **0.383** | 0.377 | 0.389 | 0.364 | 0.385 | 0.377 | 0.389 |
| | Avg. | **0.256** | **0.315** | 0.269 | 0.323 | 0.257 | 0.316 | 0.269 | 0.323 |
| | 96 | **0.131** | **0.223** | 0.134 | 0.229 | 0.132 | 0.225 | 0.147 | 0.252 |
| | 192 | **0.149** | 0.243 | 0.153 | 0.247 | 0.150 | 0.241 | 0.163 | 0.265 |
| Electricity | 336 | **0.166** | 0.260 | 0.168 | 0.264 | **0.166** | **0.258** | 0.179 | 0.280 |
| | 720 | **0.202** | **0.290** | 0.207 | 0.294 | 0.203 | **0.290** | 0.218 | 0.312 |
| | Avg. | **0.162** | 0.254 | 0.166 | 0.259 | 0.163 | **0.254** | 0.177 | 0.277 |

Table 8: Full result of the ablation study. All results are conducted on 4 different predicted window of $\{96, 192, 336, 720\}$. The best results are in **bold**.

| Metric | | **TimeDART** | | *W/o* AR | | *W/o* Diff | | *W/o* AR-Diff | |
| | | MSE | MAE | MSE | MAE | MSE | MAE | MSE | MAE |
|---|---|---|---|---|---|---|---|---|---|
| | 96 | **0.283** | **0.340** | 0.299 | 0.352 | 0.288 | 0.343 | 0.300 | 0.354 |
| | 192 | **0.345** | **0.382** | 0.364 | 0.390 | 0.351 | 0.384 | 0.365 | 0.390 |
| ETTh2 | 336 | **0.365** | **0.399** | 0.387 | 0.414 | 0.372 | 0.404 | 0.386 | 0.413 |
| | 720 | **0.390** | **0.425** | 0.409 | 0.438 | 0.396 | 0.432 | 0.406 | 0.436 |
| | Avg. | **0.346** | **0.387** | 0.365 | 0.399 | 0.352 | 0.391 | 0.364 | 0.398 |
| | 96 | **0.165** | **0.256** | 0.184 | 0.276 | 0.175 | 0.265 | 0.186 | 0.278 |
| | 192 | **0.221** | **0.294** | 0.245 | 0.317 | 0.228 | 0.300 | 0.246 | 0.318 |
| ETTm2 | 336 | **0.279** | **0.330** | 0.307 | 0.355 | 0.281 | 0.331 | 0.311 | 0.367 |
| | 720 | **0.364** | **0.385** | 0.388 | 0.403 | 0.374 | 0.392 | 0.395 | 0.420 |
| | Avg. | **0.257** | **0.316** | 0.281 | 0.338 | 0.265 | 0.322 | 0.285 | 0.346 |
| | 96 | **0.132** | **0.225** | 0.163 | 0.281 | 0.134 | 0.228 | 0.158 | 0.276 |
| | 192 | **0.150** | **0.241** | 0.179 | 0.294 | **0.150** | 0.242 | 0.163 | 0.265 |
| Electricity | 336 | **0.166** | **0.258** | 0.195 | 0.306 | 0.167 | 0.259 | 0.199 | 0.312 |
| | 720 | **0.203** | **0.290** | 0.234 | 0.335 | 0.205 | 0.292 | 0.238 | 0.341 |
| | Avg. | **0.163** | **0.254** | 0.193 | 0.304 | 0.164 | 0.255 | 0.190 | 0.299 |

In summary, both modules are indispensable for TimeDART's success. The auto-regressive mechanism is particularly important for long-term predictions, as evidenced in ETTm2, while the denoising diffusion process significantly improves accuracy and learning, especially in datasets like ETTh2.

# E  HYPERPARAMETER SENSITIVITY ANALYSIS

## E.1  HYPERPARAMETER SENSITIVITY IN FORWARD PROCESS

Table 9: Full result of hyperparameter sensitivity analysis of total noise steps and noise schedulers. All results are conducted on 4 different predicted window of $\{96, 192, 336, 720\}$. The best results are in **bold**.

| Param. | | Total Noise Steps | | | | | | Noise Scheduler | | | |
| | | 750 | | 1000 | | 1250 | | Cos. | | Lin. | |
| Metric | | MSE | MAE | MSE | MAE | MSE | MAE | MSE | MAE | MSE | MAE |
|---|---|---|---|---|---|---|---|---|---|---|---|
| | 96 | 0.288 | 0.348 | **0.283** | **0.340** | 0.285 | 0.345 | **0.283** | **0.340** | 0.294 | 0.348 |
| | 192 | 0.346 | 0.386 | **0.343** | **0.381** | 0.344 | 0.384 | **0.343** | **0.381** | 0.357 | 0.390 |
| ETTh2 | 336 | **0.364** | 0.405 | **0.364** | **0.399** | **0.364** | 0.405 | **0.364** | **0.399** | 0.375 | 0.408 |
| | 720 | 0.396 | 0.431 | **0.390** | **0.425** | 0.396 | 0.431 | **0.390** | **0.425** | 0.407 | 0.439 |
| | Avg. | 0.396 | 0.431 | **0.390** | **0.425** | 0.396 | 0.431 | **0.390** | **0.425** | 0.407 | 0.439 |
| | 96 | 0.173 | 0.265 | **0.165** | **0.256** | 0.173 | 0.265 | **0.165** | **0.256** | 0.174 | 0.263 |
| | 192 | 0.226 | 0.299 | **0.221** | **0.294** | 0.226 | 0.299 | **0.221** | **0.294** | 0.240 | 0.307 |
| ETTm2 | 336 | 0.280 | 0.333 | **0.279** | **0.330** | 0.279 | 0.333 | **0.279** | **0.330** | 0.284 | 0.334 |
| | 720 | 0.374 | 0.389 | **0.364** | **0.385** | 0.372 | 0.388 | **0.364** | **0.385** | 0.377 | 0.389 |
| | Avg. | 0.396 | 0.431 | **0.390** | **0.425** | 0.396 | 0.431 | **0.390** | **0.425** | 0.407 | 0.439 |

The results in Table 9 suggest that varying the total number of diffusion steps ($T$) has a relatively minor impact on model performance across datasets. Whether $T$ is set to 750, 1000, or 1250, the model's effectiveness remains consistent, with minimal variation in MSE values. This indicates that once a sufficient number of diffusion steps are reached, further increases offer little additional benefit.

In contrast, the noise scheduler plays a more critical role in shaping model performance. The cosine scheduler consistently outperforms the linear scheduler, with the gap in performance widening as the prediction horizon increases. For instance, in the ETTh2 dataset, the cosine scheduler shows significantly better results at longer horizons compared to the linear scheduler, highlighting its ability to facilitate smoother noise transitions. These results emphasize the importance of selecting an appropriate noise scheduler, as it greatly influences the model's ability to effectively denoise during pre-training.

## E.2  HYPERPARAMETER SENSITIVITY IN REVERSE PROCESS

The results in Table 10 indicate that increasing the number of layers in the denoising patch decoder does not consistently improve performance. While a single decoder layer generally provides the best balance between model complexity and accuracy, adding more layers tends to offer diminishing returns. In fact, beyond one or two layers, performance gains become negligible, and excessive layers can even hinder the training process by shifting capacity away from the representation network. This suggests that an overly complex decoder may underutilize the model's capacity, leading to suboptimal pre-training outcomes. Overall, the results emphasize the importance of maintaining a balanced architecture, where one decoder layer appears to be sufficient for effective performance across datasets.

Table 10: Full result of hyperparameter sensitivity analysis of the number layers of denoising patch decoder. All results are conducted on 4 different predicted window of $\{96, 192, 336, 720\}$. The best results are in **bold**.

| Numbers | | 0 | | 1 | | 2 | | 3 | |
|---|---|---|---|---|---|---|---|---|---|
| Metric | | MSE | MAE | MSE | MAE | MSE | MAE | MSE | MAE |
| ETTh2 | 96 | 0.288 | 0.343 | **0.283** | **0.340** | 0.284 | 0.345 | 0.284 | 0.345 |
| | 192 | 0.351 | 0.384 | **0.343** | **0.381** | 0.342 | 0.382 | 0.342 | 0.382 |
| | 336 | 0.372 | 0.404 | **0.364** | **0.399** | 0.360 | 0.398 | 0.361 | 0.400 |
| | 720 | 0.396 | 0.432 | **0.390** | **0.425** | 0.394 | 0.428 | 0.397 | 0.433 |
| | Avg. | 0.352 | 0.391 | **0.345** | **0.386** | **0.345** | 0.388 | 0.346 | 0.390 |
| ETTm2 | 96 | 0.175 | 0.265 | **0.165** | **0.256** | 0.166 | 0.257 | 0.167 | 0.257 |
| | 192 | 0.228 | 0.300 | **0.221** | **0.294** | 0.226 | 0.297 | 0.230 | 0.399 |
| | 336 | 0.281 | 0.331 | **0.279** | **0.330** | 0.280 | 0.333 | 0.282 | 0.337 |
| | 720 | 0.374 | 0.392 | **0.364** | **0.385** | 0.379 | 0.398 | 0.372 | 0.386 |
| | Avg. | 0.265 | 0.322 | **0.257** | **0.316** | 0.263 | 0.321 | 0.263 | 0.345 |
| Electricity | 96 | 0.134 | 0.228 | **0.132** | **0.225** | 0.134 | 0.227 | 0.142 | 0.244 |
| | 192 | **0.150** | 0.242 | **0.150** | **0.241** | 0.151 | 0.243 | 0.160 | 0.260 |
| | 336 | 0.167 | 0.259 | **0.166** | **0.258** | 0.169 | 0.258 | 0.175 | 0.274 |
| | 720 | 0.205 | 0.292 | **0.203** | **0.290** | 0.211 | 0.304 | 0.215 | 0.310 |
| | Avg. | 0.164 | 0.255 | **0.163** | **0.254** | 0.166 | 0.258 | 0.173 | 0.272 |

Table 11: Full result of hyperparameter sensitivity analysis of patch length. All results are conducted on 4 different predicted window of $\{96, 192, 336, 720\}$. The best results are in **bold**.

| Length | | 1 | | 2 | | 4 | | 8 | | 16 | |
|---|---|---|---|---|---|---|---|---|---|---|---|
| Metric | | MSE | MAE | MSE | MAE | MSE | MAE | MSE | MAE | MSE | MAE |
| ETTh2 | 96 | 0.312 | 0.364 | **0.283** | **0.340** | 0.295 | 0.348 | 0.301 | 0.356 | 0.313 | 0.365 |
| | 192 | 0.387 | 0.412 | **0.343** | **0.381** | 0.348 | 0.385 | 0.356 | 0.390 | 0.365 | 0.400 |
| | 336 | 0.419 | 0.439 | **0.364** | **0.399** | 0.369 | 0.406 | 0.370 | 0.407 | 0.377 | 0.415 |
| | 720 | 0.452 | 0.469 | **0.390** | **0.425** | 0.399 | 0.434 | 0.403 | 0.436 | 0.412 | 0.443 |
| | Avg. | 0.393 | 0.421 | **0.345** | **0.386** | 0.353 | 0.393 | 0.358 | 0.397 | 0.367 | 0.406 |
| ETTm2 | 96 | 0.169 | 0.258 | **0.165** | **0.256** | 0.177 | 0.267 | 0.168 | 0.258 | 0.170 | 0.261 |
| | 192 | 0.226 | 0.295 | **0.221** | **0.294** | 0.231 | 0.302 | 0.226 | 0.297 | 0.224 | 0.297 |
| | 336 | 0.283 | 0.333 | 0.279 | **0.330** | 0.284 | 0.336 | 0.278 | **0.330** | **0.277** | **0.330** |
| | 720 | 0.371 | 0.388 | 0.364 | 0.385 | 0.378 | 0.392 | **0.362** | **0.382** | 0.370 | 0.385 |
| | Avg. | 0.262 | 0.319 | **0.257** | **0.316** | 0.268 | 0.324 | 0.259 | 0.317 | 0.260 | 0.318 |
| Electricity | 96 | 0.165 | 0.285 | 0.149 | 0.254 | 0.135 | 0.234 | **0.132** | **0.225** | 0.146 | 0.250 |
| | 192 | 0.181 | 0.297 | 0.163 | 0.266 | 0.152 | 0.249 | **0.150** | **0.241** | 0.161 | 0.264 |
| | 336 | 0.199 | 0.312 | 0.180 | 0.282 | 0.169 | 0.266 | **0.166** | **0.258** | 0.178 | 0.281 |
| | 720 | 0.238 | 0.341 | 0.220 | 0.313 | 0.208 | 0.299 | **0.203** | **0.290** | 0.218 | 0.313 |
| | Avg. | 0.196 | 0.309 | 0.178 | 0.279 | 0.166 | 0.262 | **0.163** | **0.254** | 0.176 | 0.277 |

### E.3 HYPERPARAMETER SENSITIVITY IN INPUT PROCESS

The results in Table 11 demonstrate that patch length significantly affects model performance, with each dataset benefiting from different levels of information density. For instance, datasets like *Electricity*, which exhibit high redundancy between data points, perform best with larger patches (e.g., patch length 8), achieving the lowest average MSE of 0.163 and MAE of 0.254. In contrast, other datasets may require shorter patch lengths to capture more localized patterns. However, using smaller patches increases the computational complexity considerably, making training much more difficult and resource-intensive. Thus, determining the optimal patch length depends not only on the dataset's characteristics but also on the balance between performance and computational feasibility.

## F VISUALIZATION

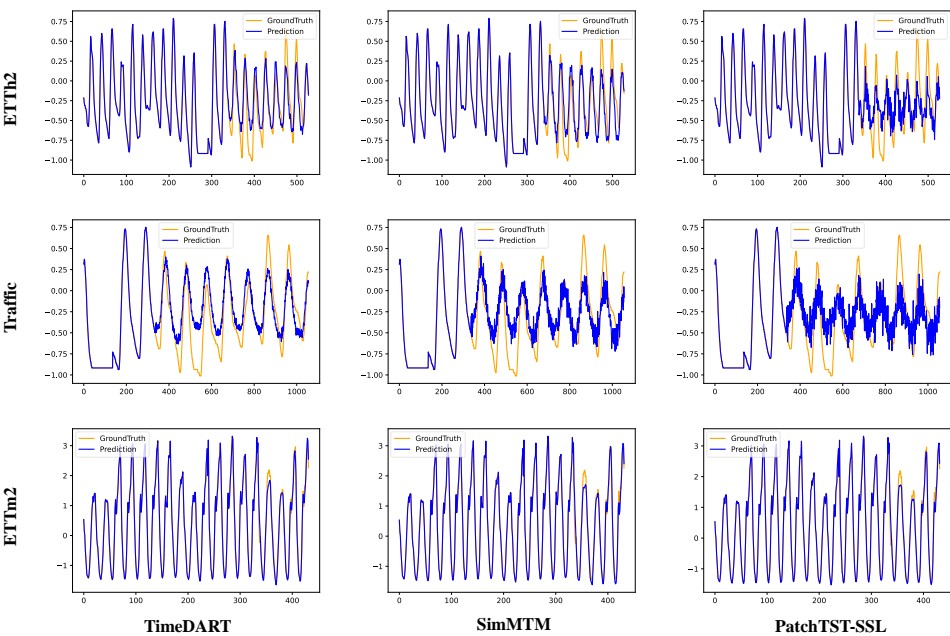

Figure 3: Illustration of forecasting showcases comparing TimeDART and baseline models. The look-back window is set to 336 and the predicted window is set to 192, 96, 720 for the ETTh2, Traffic, and ETTm2 dataset respectively.

In this visualization (Figure 3), TimeDART is compared against SimMTM and PatchTST-SSL, the self-supervised version of PatchTST. The ground truth, input data, and predictions are plotted together. The look-back window is set to 336 for all datasets, while the predicted window varies: 192 for ETTh2, 96 for Traffic, and 720 for ETTm2. This setup ensures that different datasets are forecasted over appropriate future horizons based on their unique characteristics. TimeDART consistently shows more accurate and smoother predictions, closely matching the ground truth compared to the baseline models.

