# OpenReview forum: "Diffusion Auto-regressive Transformer for Effective Self-supervised Time Series Forecasting"
_ICLR.cc/2025/Conference — Submitted to ICLR 2025_

### Official Review · Reviewer_gMoC · 2024-10-23

**Soundness:** 1
**Presentation:** 2
**Contribution:** 2
**Rating:** 3
**Confidence:** 5

**Summary:**

This paper proposes TimeDART, a novel self-supervised learning paradigm that hopes to learn transferable generic representations from unlabeled data through pre-training and then fine-tune them to different downstream tasks, e.g., forecasting tasks, etc. In this paper, a unified self-supervised learning framework is established by fusing the diffusion-denoising process and autoregressive modeling. By learning global sequence dependencies and local detail features in multi-domain time series data, the model's ability to capture comprehensive time series features and cross-domain generalization ability are improved. Specifically, TimeDART designs a cross-attention based denoising decoder in the diffusion mechanism, which improves the effectiveness of time series pre-training by significantly enhancing the model's capability to capture features within local blocks.

**Strengths:**

The authors outline the main approaches to self-supervised learning in the time series domain, including mask reconstruction and contrast learning, and analyses the shortcomings of the two existing self-supervised learning paradigms separately, e.g., the mask reconstruction-based approach introduces a huge gap between pre-training and fine-tuning, and the contrast-learning-based approach prioritizes the capture of discriminative features, which lead to a huge discrepancy between pre-training tasks and fine-tuning tasks.

In addition, the authors raise two issues critical to the self-training paradigm, 1) how to narrow the gap between the pre-training target and the downstream fine-tuning task, and 2) modelling both long-term dependencies and local pattern information in the self-supervised pre-training phase. However, we believe that TimeDART is not the best solution to these problems.

**Weaknesses:**

**Weakness 1:**

As a Diffusion-based model, we would like to introduce more valuable metrics to fully examine the performance of the proposed TimeDART. Specifically, in order to evaluate the prediction accuracy and generalization capability of TimeDART from both forecasting and generation perspectives, existing studies often include the following four metrics (including Context-FID, Correlational Score, Discriminative Score, and Predictive Score) for comprehensively evaluating the performance of Diffusion methods.

In addition, the proposed TimeDART is not compared with advanced Diffusion-based approaches, such as Diffusion-TS[1], mr-Diff[2], and MG-TSD[3]. We believe that the introduction of more competitive and up-to-date approaches can demonstrate the effectiveness of the proposed method more objectively.

**Weakness 2:**

In the in-domain setting (Table 2), there are the following weaknesses:

* The performance improvement of the proposed TimeDART over SOTA methodologies SimMTM and PatchTST is less than 5%, which indicates that the performance improvement of the model is not obvious.

* In addition, we note that the performance of "Random Init" shown in Table 1 is slightly worse than Supervised PatchTST. Does this indicate that in the supervised setting of a single domain, the proposed model architecture exhibits worse performance compared to PatchTST? If the modeling capability of the TimeDART is poor in small datasets, the model will often show worse generalization ability in cross-domain pre-training, which leads to the rationality of the model architecture being questioned.

**Weakness 3:**

In the cross-domain setting (Table 3), there are the following weaknesses that can be improved:

* The model is pre-trained on only two power domain datasets (ETT and Electricity). As a result, only the single domain information is included in the model, which limits the generalization ability of the model under cross-domain challenges. Recent unified time series forecasting models include two paradigms. The first is unified models based on LLM fine-tuning, such as OneFitsAll[4] and TimeLLM[5]. This is followed by pre-training on a multi-domain hybrid Time series dataset followed by fine-tuning on specific downstream tasks, e.g., Timer[6], Moriai[7] and MOMENT[8]. In conclusion, we believe that introducing information from more domains during pretraining can improve the cross-domain generalization of the model, and TimeDART is expected to be pretrained on a wider range of datasets.

* Table 3 only shows the performance comparison of TimeDART under different Settings; however, it lacks the comparison with the latest baseline. In fact, recent UniTime[9] have achieved joint pretraining across multiple domains. Therefore, we expect the authors to introduce more advanced baselines to compare with TimeDART, which will help us get a comprehensive understanding of TimeDART's performance.

**Weakness 4:**

In ablation experiments (Table 4), there are the following drawbacks:

* There is a lack of detailed descriptions specific to ablation experiments, such as in-domain Settings or cross-domain Settings. When we compare the results in Table 4 and Table 2, we can speculate that the ablation experiment is only carried out in the in-domain setting. However, in the cross-domain setting, the reader is eager to know whether the proposed autoregressive diffusion model is effective.

* In Table 4," The "W/o AR" model obtained the improved "W/o AR" model after introducing the Diffusion-based decoder; However, the performance of the latter was slightly degraded on the ETTh2 and Electricity datasets. This may indicate that the predictions of the model become worse when the diffusion model is introduced. This casts doubt on the rationality of introducing a Diffusion-Denoising process in the Decoder.

**Reference:**

1) Yuan, Xinyu and Yan Qiao. “Diffusion-TS: Interpretable Diffusion for General Time Series Generation.” ICLR 2024.

2) Shen, Lifeng et al. “Multi-Resolution Diffusion Models for Time Series Forecasting.” ICLR 2024.

3) Fan, Xinyao et al. “MG-TSD: Multi-Granularity Time Series Diffusion Models with Guided Learning Process.”  ICLR 2024.

4) Zhou, Tian et al. “One Fits All: Power General Time Series Analysis by Pretrained LM.” NIPS 2023.

5) Jin, Ming et al. “Time-LLM: Time Series Forecasting by Reprogramming Large Language Models.” ICLR 2024.

6) Liu, Yong et al. “Timer: Generative Pre-trained Transformers Are Large Time Series Models.” ICML 2024.

7) Woo, Gerald et al. “Unified Training of Universal Time Series Forecasting Transformers.” ICML 2024.

8) Goswami, Mononito et al. “MOMENT: A Family of Open Time-series Foundation Models.” ICML 2024.

9) Liu, Xu et al. “UniTime: A Language-Empowered Unified Model for Cross-Domain Time Series Forecasting.” Proceedings of the ACM on Web Conference 2024.

**Questions:**

**Question 1:**

This paper uses an autoregressive prediction paradigm, however autoregressive approaches generally suffer from error accumulation and high inference time overheads. To the best of our knowledge, TimeDART has not designed a special training-inference strategy or model architecture to mitigate these problems. Further discussion of TimeDART's limitations in the autoregressive prediction paradigm is urgent and necessary.

**Question 2:**

As the authors say ‘we adopted the channel-independence setting’. However, for datasets with a very large number of channels, such as ECL and Traffic, how can a channel-independent design establish connections between multiple channels? Meanwhile, the experimental results of Crossformer[10], SAMformer[11] and iTransformer[12], show that cross-channel modelling is crucial for multivariate time series, and we believe that a discussion on inter-channel modelling for multivariate time series is necessary. However, to the best of our knowledge, TimeDART does not consider these factors in its modelling.

**Question 3:**

See Weakness 2,3 for our concerns about the experimental results, and we believe that the introduction of competitive and up-to-date baselines is considered necessary.  In addition, the authors are expected to explain the phenomena in the ablation experiments, details of which can be referred to Weakness 4.

**Reference:**

10) Zhang, Yunhao and Junchi Yan. “Crossformer: Transformer Utilizing Cross-Dimension Dependency for Multivariate Time Series Forecasting.” ICLR 2023.

11) Ilbert, Romain et al. “SAMformer: Unlocking the Potential of Transformers in Time Series Forecasting with Sharpness-Aware Minimization and Channel-Wise Attention.” ICML 2024.

12) Liu, Yong et al. “iTransformer: Inverted Transformers Are Effective for Time Series Forecasting.” ICLR 2024.

---

> ### Author Response · Authors · 2024-11-19
>
> **Q1:** As a Diffusion-based model, we would like to introduce more valuable metrics to fully examine the performance of the proposed TimeDART. Specifically, in order to evaluate the prediction accuracy and generalization capability of TimeDART from both forecasting and generation perspectives, existing studies often include the following four metrics (including Context-FID, Correlational Score, Discriminative Score, and Predictive Score) for comprehensively evaluating the performance of Diffusion methods.
> In addition, the proposed TimeDART is not compared with advanced Diffusion-based approaches, such as Diffusion-TS[1], mr-Diff[2], and MG-TSD[3]. We believe that the introduction of more competitive and up-to-date approaches can demonstrate the effectiveness of the proposed method more objectively.
>
> **A1:** We have noted your concerns regarding the evaluation metrics and the comparison with other Diffusion-based approaches. However, it seems there is a significant misunderstanding about our framework, which we would like to address and clarify.
>
> Unlike traditional Diffusion-based models that utilize the denoising process for both training and inference, TimeDART employs the Diffusion Model exclusively for self-supervised pretraining. In our framework, the denoising process is designed to optimize the representation network, which is then transferred to downstream tasks. Importantly, the denoising network itself is not used during forecasting. Instead, we adopt a simple yet effective one-step prediction scheme for downstream tasks. This design allows us to avoid issues such as error accumulation in auto-regressive settings and the computational overhead of the reverse diffusion process.
>
> Given this structure, metrics such as Context-FID, Correlational Score, Discriminative Score, and Predictive Score—which are typically employed to evaluate the generative capabilities of Diffusion-based models—are irrelevant for assessing TimeDART's downstream performance. Our focus is on leveraging the pretraining benefits of the Diffusion Model, not on direct generative forecasting, and thus these metrics do not align with the objectives of our method.
>
> Regarding your suggestion to compare our model with advanced Diffusion-based approaches like Diffusion-TS, mr-Diff, and MG-TSD, it is crucial to note that our method operates in a fundamentally different paradigm. These methods rely heavily on the denoising process for direct probabilistic forecasting, leading to substantial computational and inference costs. Moreover, their performance on long-term, multi-variable forecasting tasks is often subpar. To provide a fair and concrete comparison, we specifically selected an open-source baseline, Diffusion-TS, for evaluation. The results under comparable settings, summarized below, highlight its inefficiency and inferior accuracy, further substantiating the advantages of our approach:
>
> | Models | Metrics | TimeDART MSE | TimeDART MAE | Random init MSE | Random init MAE | Diffusion-TS MSE | Diffusion-TS MAE |
> | ------ | ------- | ------------ | ------------ | --------------- | --------------- | ---------------- | ---------------- |
> | ETTh2  | 96      | **0.283**    | **0.340**    | 0.294           | 0.348           | 1.772            | 1.130            |
> |        | 192     | **0.345**    | **0.382**    | 0.357           | 0.390           | 1.781            | 1.154            |
> |        | 336     | **0.365**    | **0.399**    | 0.375           | 0.408           | 1.794            | 1.167            |
> |        | 720     | **0.390**    | **0.425**    | 0.407           | 0.439           | 1.806            | 1.182            |
> |        | Avg.    | **0.346**    | **0.387**    | 0.358           | 0.396           | 1.788            | 1.158            |
> | ETTm2  | 96      | **0.165**    | **0.256**    | 0.174           | 0.263           | 1.656            | 1.074            |
> |        | 192     | **0.221**    | **0.294**    | 0.240           | 0.307           | 1.674            | 1.082            |
> |        | 336     | **0.279**    | **0.330**    | 0.284           | 0.334           | 1.682            | 1.091            |
> |        | 720     | **0.364**    | **0.385**    | 0.377           | 0.389           | 1.695            | 1.096            |
> |        | Avg.    | **0.257**    | **0.316**    | 0.269           | 0.323           | 1.677            | 1.086            |
>
> It is also important to note that most diffusion-based models focus on probabilistic forecasting and are evaluated on metrics like Context-FID, as you mentioned. Consequently, their performance on point-level evaluations, such as those in our study, tends to be significantly worse, resulting in a considerable gap compared to our method.
>
> We hope this explanation resolves any misunderstandings and provides a clearer perspective on the contributions of our work.

---

> > ### Author Response · Authors · 2024-11-19
> >
> > **Q2:** In the in-domain setting (Table 2), there are the following weaknesses:
> > - The performance improvement of the proposed TimeDART over SOTA methodologies SimMTM and PatchTST is less than 5%, which indicates that the performance improvement of the model is not obvious.
> > - In addition, we note that the performance of "Random Init" shown in Table 1 is slightly worse than Supervised PatchTST. Does this indicate that in the supervised setting of a single domain, the proposed model architecture exhibits worse performance compared to PatchTST? If the modeling capability of the TimeDART is poor in small datasets, the model will often show worse generalization ability in cross-domain pre-training, which leads to the rationality of the model architecture being questioned.
> >
> > **A2:** We sincerely thank the reviewer for their detailed observations and thoughtful comments regarding the in-domain performance of TimeDART.
> >
> > 1. **Regarding the performance improvement of TimeDART over SOTA models like PatchTST and SimMTM:**
> >
> >    The relatively small performance improvement of less than 5% is indeed a valid concern. However, it is important to note that significant performance improvements in time series forecasting, particularly in comparison to state-of-the-art (SOTA) methods, are rare in the field. For example, several recent works in the domain, such as TimeLLM and UniTime, report performance improvements of only 1% to 2% on certain datasets like ETTh1, ETTh2, and Weather. This highlights that incremental improvements of 5% are considered notable in this domain. Moreover, we have observed that our model performs as SOTA in most cases, and even when it doesn't, it generally ranks as the second-best. We believe that the improvements shown by TimeDART are meaningful and indicate strong performance, particularly when considering the high baseline performance of existing models like SimMTM and PatchTST.
> >
> > 2. **Regarding the performance of "Random Init" being slightly worse than supervised PatchTST:**
> >
> >    The Random Init performance being slightly worse than Supervised PatchTST is indeed a point worth addressing. The first key difference lies in the patch length and overlap setting. PatchTST uses patch length = 16 and an overlap of 8, meaning that each patch overlaps with the next one, allowing for cross-patch information flow. In contrast, to ensure the proper functioning of the auto-regressive mechanism in TimeDART, we do not allow overlap between patches, which introduces a fundamental difference in the way information is encoded.
> >
> >    Additionally, PatchTST is trained in a fully supervised manner for 100 epochs, while for the Random Init setting in TimeDART, we trained the model for only 10 epochs to remain consistent with the pretraining phase and other baselines. This difference in training duration could explain the slight performance gap, as the PatchTST model has been trained more extensively. Therefore, the Random Init setting in TimeDART is expected to show somewhat lower performance compared to supervised PatchTST due to the shorter training period and the design of the patching mechanism.

---

> > > ### Author Response · Authors · 2024-11-19
> > >
> > > **Q3:** In the cross-domain setting (Table 3), there are the following weaknesses that can be improved:
> > > - The model is pre-trained on only two power domain datasets (ETT and Electricity). As a result, only the single domain information is included in the model, which limits the generalization ability of the model under cross-domain challenges. Recent unified time series forecasting models include two paradigms. The first is unified models based on LLM fine-tuning, such as OneFitsAll[4] and TimeLLM[5]. This is followed by pre-training on a multi-domain hybrid Time series dataset followed by fine-tuning on specific downstream tasks, e.g., Timer[6], Moriai[7] and MOMENT[8]. In conclusion, we believe that introducing information from more domains during pretraining can improve the cross-domain generalization of the model, and TimeDART is expected to be pretrained on a wider range of datasets.
> > > - Table 3 only shows the performance comparison of TimeDART under different Settings; however, it lacks the comparison with the latest baseline. In fact, recent UniTime[9] have achieved joint pretraining across multiple domains. Therefore, we expect the authors to introduce more advanced baselines to compare with TimeDART, which will help us get a comprehensive understanding of TimeDART's performance.
> > >
> > > **A3:** We sincerely thank the reviewer for their insightful comments regarding the cross-domain generalization of TimeDART and the inclusion of more advanced baselines.
> > >
> > > Our initial motivation for pretraining on five energy system domain datasets was to validate the model’s generalization ability within the same domain and across different datasets. However, we fully agree with your suggestion that expanding the pretraining to include a broader range of domains can significantly enhance the model’s cross-domain adaptability. In response, we worked tirelessly, often through the night, to conduct additional experiments. Specifically, we performed general pretraining on eight diverse datasets, followed by fine-tuning on specific downstream tasks. This expanded setup includes datasets from both energy systems and other domains, enabling a more comprehensive evaluation of the model's cross-domain performance.
> > >
> > > Additionally, we compared TimeDART with advanced baselines, including UniTime, GPT4TS, and PatchTST. Due to GPU resource limitations, we could not replicate UniTime’s results directly. Instead, we followed UniTime’s experimental setup, which includes a look-back window of 96 and predicted windows of {96, 192, 336, 720}, and used their reported results for comparison.
> > >
> > > As of now, we have completed fine-tuning experiments on ETTh2, ETTm2, Exchange, and Electricity datasets during the rebuttal phase. The results, summarized in the **table1** below, demonstrate the potential of TimeDART to serve as a unified foundation model capable of performing well across various domains.
> > >
> > > After pretraining on all datasets and fine-tuning on specific ones, TimeDART significantly outperforms models including UniTime. Additionally, our model is more lightweight, with shorter training and fine-tuning times.
> > >
> > > **Q4:** The model uses a denoising diffusion loss, which may not be the most suitable for every type of forecasting task. How does TimeDART perform with other loss functions (e.g., Quantile Loss, Huber Loss) that are often used in time series forecasting tasks where the goal is to forecast confidence intervals or robustly handle outliers?
> > >
> > > **A4:** We sincerely thank the reviewer for raising this insightful question. Our response is structured as follows:
> > > - **Clarification of Diffusion Loss Usage**
> > >
> > >    We would like to clarify that the diffusion loss is used exclusively during the self-supervised pretraining phase, not in downstream tasks. Its purpose is to model the underlying data distribution via a denoising process, which helps TimeDART capture richer representations. While the diffusion loss has a form similar to MSE, it is fundamentally different from traditional forecasting losses like Quantile Loss and Huber Loss. Unlike these losses, which extend MSE or MAE for specific objectives, the diffusion loss is designed to improve the model’s ability to generate diverse, high-quality predictions for complex forecasting tasks.
> > > - **MSE in Downstream Tasks**
> > >
> > >    In downstream forecasting tasks, we follow the standard practice of using MSE as the optimization objective, which aligns with the majority of self-supervised methods. To ensure a fair comparison, all baselines in our experiments also use MSE as their loss function. This allows us to isolate and evaluate the benefits of the pretrained representations.
> > > We acknowledge the reviewer’s suggestion to explore alternative loss functions, such as Quantile Loss and Huber Loss, for downstream tasks. This is indeed a valuable direction, and we plan to extend our experiments in future work to assess the impact of these losses on forecasting performance.

---

> > > > ### Author Response · Authors · 2024-11-19
> > > > **Table1: Comparison on cross-domain setting. Model is trained on 8 datasets and finetuned on a single dataset.**
> > > >
> > > > | Models             | Metrics | TimeDART (MSE) | TimeDART (MAE) | Random init (MSE) | Random init (MAE) | Unitime (MSE) | Unitime (MAE) | GPT4TS (MSE) | GPT4TS (MAE) | PatchTST (MSE) | PatchTST (MAE) |
> > > > | ------------------ | ------- | -------------- | -------------- | ----------------- | ----------------- | :------------ | ------------- | ------------ | ------------ | -------------- | -------------- |
> > > > | ALL -> ETTh2       | 96      | **0.293**      | **0.339**      | 0.296             | 0.342             | 0.296         | 0.345         | 0.303        | 0.349        | 0.314          | 0.361          |
> > > > |                    | 192     | **0.374**      | **0.390**      | 0.382             | 0.394             | **0.374**     | 0.394         | 0.391        | 0.399        | 0.407          | 0.411          |
> > > > |                    | 336     | **0.410**      | **0.419**      | 0.422             | 0.430             | 0.415         | 0.427         | 0.429        | 0.449        | 0.437          | 0.443          |
> > > > |                    | 720     | **0.425**      | **0.444**      | 0.436             | 0.449             | **0.425**     | **0.444**     | 0.430        | 0.449        | 0.434          | 0.448          |
> > > > |                    | Avg.    | **0.376**      | **0.398**      | 0.384             | 0.404             | 0.378         | 0.403         | 0.386        | 0.406        | 0.398          | 0.416          |
> > > > | ALL -> ETTm2       | 96      | **0.180**      | 0.270          | 0.195             | 0.289             | 0.183         | **0.266**     | 0.229        | 0.304        | 0.240          | 0.318          |
> > > > |                    | 192     | **0.245**      | **0.305**      | 0.267             | 0.333             | 0.251         | 0.310         | 0.287        | 0.338        | 0.301          | 0.352          |
> > > > |                    | 336     | **0.308**      | **0.348**      | 0.311             | 0.355             | 0.319         | 0.351         | 0.337        | 0.367        | 0.367          | 0.391          |
> > > > |                    | 720     | **0.413**      | **0.409**      | 0.431             | 0.421             | 0.420         | 0.410         | 0.430        | 0.416        | 0.451          | 0.432          |
> > > > |                    | Avg.    | **0.287**      | **0.333**      | 0.301             | 0.350             | 0.293         | 0.334         | 0.321        | 0.356        | 0.340          | 0.373          |
> > > > | ALL -> Exchange    | 96      | **0.082**      | 0.211          | 0.094             | 0.212             | 0.086         | **0.209**     | 0.142        | 0.261        | 0.137          | 0.260          |
> > > > |                    | 192     | **0.172**      | **0.299**      | 0.212             | 0.332             | 0.174         | **0.299**     | 0.224        | 0.339        | 0.222          | 0.341          |
> > > > |                    | 336     | 0.329          | 0.418          | 0.365             | 0.442             | **0.319**     | **0.408**     | 0.377        | 0.448        | 0.372          | 0.447          |
> > > > |                    | 720     | **0.861**      | **0.697**      | 0.886             | 0.709             | 0.875         | 0.701         | 0.939        | 0.736        | 0.912          | 0.727          |
> > > > |                    | Avg.    | **0.361**      | 0.406          | 0.389             | 0.424             | 0.364         | **0.404**     | 0.421        | 0.446        | 0.411          | 0.444          |
> > > > | ALL -> Electricity | 96      | **0.178**      | **0.269**      | 0.189             | 0.287             | 0.189         | 0.287         | 0.198        | 0.290        | 0.202          | 0.293          |
> > > > |                    | 192     | **0.182**      | **0.273**      | 0.200             | 0.290             | 0.199         | 0.291         | 0.234        | 0.325        | 0.223          | 0.318          |
> > > > |                    | 336     | **0.199**      | **0.297**      | 0.206             | 0.301             | 0.214         | 0.305         | 0.249        | 0.338        | 0.223          | 0.318          |
> > > > |                    | 720     | **0.241**      | **0.332**      | 0.251             | 0.333             | 0.254         | 0.335         | 0.289        | 0.366        | 0.259          | 0.341          |
> > > > |                    | Avg.    | **0.200**      | **0.293**      | 0.212             | 0.305             | 0.216         | 0.305         | 0.251        | 0.338        | 0.221          | 0.311          |

---

> ### Author Response · Authors · 2024-11-19
>
> **Q5:** This paper uses an autoregressive prediction paradigm, however autoregressive approaches generally suffer from error accumulation and high inference time overheads. To the best of our knowledge, TimeDART has not designed a special training-inference strategy or model architecture to mitigate these problems. Further discussion of TimeDART's limitations in the autoregressive prediction paradigm is urgent and necessary.
>
> **A5:** We would like to clarify that our use of the autoregressive mechanism is limited to the self-supervised pretraining phase. During this phase, we leverage the Diffusion Model to capture complex patterns and generate robust representations. However, when transitioning to the downstream forecasting tasks, we do not reuse the denoising network for prediction. Instead, we adopt a traditional one-step prediction approach, which avoids the issues typically associated with auto-regressive error accumulation and the uncertainty introduced by the reverse sampling process in diffusion models.
>
> **Q6:** As the authors say 'we adopted the channel-independence setting'. However, for datasets with a very large number of channels, such as ECL and Traffic, how can a channel-independent design establish connections between multiple channels? Meanwhile, the experimental results of Crossformer[10], SAMformer[11] and iTransformer[12], show that cross-channel modelling is crucial for multivariate time series, and we believe that a discussion on inter-channel modelling for multivariate time series is necessary. However, to the best of our knowledge, TimeDART does not consider these factors in its modelling.
>
> **A6:** We appreciate the reviewer’s insightful comment regarding the importance of cross-channel modeling for multivariate time series.
>
> We acknowledge that both channel-independent (CI) and channel-dependent (CD) strategies have their respective strengths. Channel-dependent models, as demonstrated by works such as Crossformer, SAMformer, and iTransformer, can effectively capture inter-channel dependencies, thereby leveraging richer information for improved performance in certain tasks. However, CI models offer distinct advantages in terms of robustness and generalization, particularly in cross-domain scenarios.
>
> Recent research, such as [1] highlights the trade-offs between these two strategies. The study shows that while CD approaches may exhibit higher capacity by modeling inter-channel dependencies, the CI strategy achieves superior robustness, especially in handling non-stationary and diverse time series data. Specifically, the CI approach minimizes the risk of overfitting to domain-specific inter-channel relationships, making it better suited for real-world applications where cross-domain generalization is critical.
>
> We acknowledge that inter-channel modeling is important, as you have rightly pointed out. Cross-domain performance is indeed critical for evaluating the generalization and robustness of time series models. Given this, we adopted the channel-independent (CI) strategy, which is currently one of the simplest and most effective paradigms for conducting cross-domain experiments. CI allows us to avoid overfitting to specific inter-channel relationships, thereby enhancing the model’s robustness and adaptability across diverse datasets. This design choice aligns with our objective of building a model capable of generalizing well across domains.
>
> [1] Lu Han et al. "The Capacity and Robustness Trade-off: Revisiting the Channel Independent Strategy for Multivariate Time Series Forecasting" 2023
>
> **Q7:** See Weakness 2,3 for our concerns about the experimental results, and we believe that the introduction of competitive and up-to-date baselines is considered necessary. In addition, the authors are expected to explain the phenomena in the ablation experiments, details of which can be referred to Weakness 4.
>
> **A7:** See **A2**, **A3** and **A4**.

---

> ### Comment · Reviewer_gMoC · 2024-11-26
>
> Thanks for your rebuttal. However, I still have some concerns.
>
> 1. The authors mention that TimeDART has a unique prediction paradigm (not relying on noise reduction networks), so additional metrics do not apply to this model. However, we believe it is necessary to discuss TimeDART's performance in unsupervised generation of time series tasks, such as DiffusionTS, which supports both unsupervised generation tasks and supervised prediction tasks. We therefore believe that validating TimeDART's performance in generative tasks using four additional metrics will help to paint a comprehensive picture of TimeDART's generative capabilities.
>
> 2. Regarding the performance of ‘Random Init’ being slightly worse than supervised PatchTST, we believe that the difference in message encoding and tokeniser operation alone does not lead to such a significant performance gap. The authors should further explore the underlying reasons, such as ensuring sufficient training time.
>
> 3. The performance improvement in cross-domain training is not significant, in Table 1 of the authors‘ response, TimeDART only improves 0.002, 0.004, 0.003 on ETTh2, ETTm2, and Exchange compared to UniTime, which is not in line with the authors’ description of the performance ‘TimeDART significantly outperforms models including UniTime’. Does this mean that TimeDART lacks sufficient cross-domain generalisation capabilities?

---

> > ### Author Response · Authors · 2024-11-27
> >
> > Dear Reviewer:
> >
> > Let us now address the concerns you raised based on the above rebuttal:
> >
> > **Q1**: The authors mention that TimeDART has a unique prediction paradigm (not relying on noise reduction networks), so additional metrics do not apply to this model. However, we believe it is necessary to discuss TimeDART's performance in unsupervised generation of time series tasks, such as DiffusionTS, which supports both unsupervised generation tasks and supervised prediction tasks. We therefore believe that validating TimeDART's performance in generative tasks using four additional metrics will help to paint a comprehensive picture of TimeDART's generative capabilities.
> >
> > **A1**: Thank you for suggesting additional metrics to evaluate TimeDART's generative performance. However, there appears to be a misunderstanding regarding our architecture. TimeDART is not designed as a generative model for time series, nor is it a fully generative framework. Instead, it integrates auto-regressive and diffusion-based techniques as part of a self-supervised optimization strategy, with a specific focus on point forecasting, rather than probabilistic forecasting. This distinction in prediction paradigms is a key difference between our approach and existing diffusion-based methods. For downstream tasks, we transfer the representation network alone, without utilizing the probabilistic sampling process inherent in diffusion models for data generation. Consequently, the proposed metrics are not applicable for evaluating TimeDART's overall performance.
> >
> > **Q2**: Regarding the performance of ‘Random Init’ being slightly worse than supervised PatchTST, we believe that the difference in message encoding and tokeniser operation alone does not lead to such a significant performance gap. The authors should further explore the underlying reasons, such as ensuring sufficient training time.
> >
> > **A2**: In fact, on datasets like ETTm1 and Traffic, the performance of "Random Init" exceeds that of the supervised version of PatchTST, and on datasets like ETTh2 and ETTm2, the results are largely comparable.
> >
> > We believe that the length of the patch and whether it overlaps are crucial for data representation (as discussed in our parameter sensitivity experiments). Additionally, for PatchTST, we retained all model-specific parameters (e.g., model dimensions, patch len, etc.) from the official PatchTST implementation to ensure consistency. However, the parameters of the random initialization model were aligned with those of our TimeDART model. This alignment was necessary to fairly evaluate the effectiveness of self-supervised pretraining. As a result, there are differences in parameter settings compared to the supervised version of PatchTST, which naturally leads to variations in performance.
> >
> > **Q3**: The performance improvement in cross-domain training is not significant, in Table 1 of the authors‘ response, TimeDART only improves 0.002, 0.004, 0.003 on ETTh2, ETTm2, and Exchange compared to UniTime, which is not in line with the authors’ description of the performance ‘TimeDART significantly outperforms models including UniTime’. Does this mean that TimeDART lacks sufficient cross-domain generalisation capabilities?
> >
> > **A3**: While we acknowledge that the improvement trend is not significant in certain cases, such as for shorter prediction lengths, TimeDART demonstrates clear advantages in many other scenarios. For instance, it achieves more noticeable improvements over UniTime on longer horizons (e.g., forecasting lengths of 336 and 720) and larger datasets (e.g., Electricity), showcasing its effectiveness. Furthermore, even without adopting a large backbone model with substantial prior knowledge, such as GPT-2, or incorporating domain-specific knowledge as UniTime does, TimeDART still outperforms UniTime. This highlights the robustness and effectiveness of our approach in enhancing cross-domain performance.

---

### Official Review · Reviewer_YroL · 2024-10-27

**Soundness:** 3
**Presentation:** 3
**Contribution:** 2
**Rating:** 5
**Confidence:** 3

**Summary:**

To effectively capture both the global sequence dependence and local detail features within time series data, this paper proposed a novel generative self-supervised method called TimeDART, denoting Diffusion Auto-regressive Transformer for Time series forecasting. Extensive experiments demonstrate that TimeDART achieves state-of-the-art fine-tuning performance compared to the most advanced competitive methods in forecasting tasks.

**Strengths:**

1. Well-written
2. The author provides the code
3. The performance of the model is proved by experiments

**Weaknesses:**

1. In the original paper [1], the performance of patchTST seems to be better, and it is suggested that the author should evaluate it more fairly.
2.  Why didn't the author consider evaluating the performance of classification tasks? Currently, aside from forecasting tasks, most self-supervised learning models [2] also focus on the performance of classification tasks. This is because the main purpose of self-supervised learning is to enhance the quality of representations generated by the model and to uncover key semantic features, which is especially important for classification tasks.
3.  A crucial role of self-supervised learning is to improve the performance of backbone models [3], but the author only uses Transformer as the backbone. I am curious whether the proposed method would still be effective if MLP or TCN were used as the backbone.
4. The author's core motivation remains to capture both global and local dependencies, which is similar to most existing works [1] [4]. In other words, this paper lacks a central challenging problem, making the contribution at the motivational level somewhat limited.
5. Considering that the core motivation of this paper is to capture global and local dependencies, I suggest the author evaluate the model's performance on datasets with stable patterns, such as PEMS04. This is because datasets like ETT and Exchange have inherent issues with distributional shifts [5].

[1] A time series is worth 64 words: Long-term forecasting with transformers

[2] SimMTM: A Simple Pre-Training Framework for Masked Time-Series Modeling

[3]  Cost: Contrastive learning of disentangled seasonal-trend representations for time series forecasting

[4] Segrnn: Segment recurrent neural network for long-term time series forecasting

[5] Exploring Progress in Multivariate Time Series Forecasting: Comprehensive Benchmarking and Heterogeneity Analysis

**Questions:**

See weaknesses

---

> ### Author Response · Authors · 2024-11-19
>
> **Q1:** In the original paper [1], the performance of patchTST seems to be better, and it is suggested that the author should evaluate it more fairly.
>
> **A1:** We sincerely thank the reviewer for their insightful comment regarding the performance comparison with PatchTST.
> To address this, we would like to emphasize that all our experiments were conducted using the official implementations from the baseline repositories, and the results for PatchTST were obtained through our own runs. To ensure a fair comparison, we maintained consistency in key hyperparameters such as the **learning rate schedule** and **dropout rate** across all models.
> It is important to note that the official PatchTST implementation performs additional fine-tuning of these parameters, which we believe is not a fair comparison, given the significant computational overhead required for such optimizations. As a result, the performance we observed for PatchTST was slightly lower than the results reported in the original paper.
>
> **Q2:** Why didn't the author consider evaluating the performance of classification tasks? Currently, aside from forecasting tasks, most self-supervised learning models [2] also focus on the performance of classification tasks. This is because the main purpose of self-supervised learning is to enhance the quality of representations generated by the model and to uncover key semantic features, which is especially important for classification tasks.
>
> **A2:** We sincerely thank the reviewer for raising this insightful question regarding the evaluation of self-supervised learning models on downstream classification tasks.
> Our initial motivation was to address forecasting tasks by capturing the causal characteristics of time series data while minimizing the gap between pretraining and downstream tasks. To achieve this, we combined two generative paradigms—auto-regressive mechanisms and the Diffusion Model—within a self-supervised framework. This design choice was driven by the nature of forecasting tasks, which benefit from leveraging both causal dependencies and robust data representations.
> That said, we fully agree with your observation that self-supervised learning should aim to produce high-quality, versatile representations applicable to a wide range of downstream tasks, including classification. In response, we worked tirelessly to conduct preliminary classification experiments during the rebuttal phase. Specifically, we evaluated our model on the HAR and EEG datasets. The results are summarized below:
>
> | Compared Models       | EEG Accuracy | EEG F1 Score | HAR Accuracy | HAR F1 Score |
> | --------------------- | ------------ | ------------ | ------------ | ------------ |
> | PatchTST (supervised) | 0.8076       | 0.5460       | 0.8738       | 0.8773       |
> | SimMTM                | 0.8165       | **0.6123**   | 0.9200       | 0.9220       |
> | TimeMAE               | 0.8248       | 0.5865       | 0.9204       | **0.9284**   |
> | TimeDART Random init  | 0.7868       | 0.5138       | 0.8931       | 0.8946       |
> | TimeDART              | **0.8269**   | 0.5983       | **0.9247**   | 0.9249       |
>
> These results show that our self-supervised framework achieves competitive performance on classification tasks. Notably, TimeDART outperforms its random initialization counterpart, demonstrating the effectiveness of our pretraining in improving representation quality. On the HAR and EEG dataset, TimeDART achieves the highest accuracy, underscoring its robustness in handling diverse classification tasks.
> Due to time constraints, we have only been able to complete experiments on these datasets so far. However, if we are fortunate enough to be accepted, we will extend our evaluation to a broader range of datasets and tasks to further validate the generalizability and robustness of our approach.

---

> > ### Author Response · Authors · 2024-11-19
> >
> > **Q3:** A crucial role of self-supervised learning is to improve the performance of backbone models [3], but the author only uses Transformer as the backbone. I am curious whether the proposed method would still be effective if MLP or TCN were used as the backbone.
> >
> > **A3:** We sincerely thank the reviewer for their insightful question regarding the choice of backbone models in self-supervised learning. Your understanding of the importance of evaluating different backbones is deeply appreciated. Below, we address this concern in detail:
> > - **Why We Use Transformer?**
> > Our pretraining framework leverages an auto-regressive mechanism with a causal mask to model sequential data effectively. The Transformer backbone is particularly well-suited for this task due to its ability to capture complex dependencies over long sequences. Additionally, most state-of-the-art methods in this domain utilize Transformers as their backbone, making it a natural and competitive choice for our framework.
> > - **MLP as Backbone**
> > We explored the possibility of using MLP as the backbone but found that it is not compatible with our self-supervised framework. The MLP architecture struggles to implement the causal mask and auto-regressive mechanisms effectively, which are critical to our pretraining approach. To validate this, we compared TimeDART with DLinear, a method that employs MLP as its backbone. Our results consistently show that TimeDART outperforms DLinear across most tasks.
> > - **TCN as Backbone**
> > Temporal Convolutional Networks (TCNs), particularly causal TCNs, are indeed compatible with our framework. To investigate their potential, we replaced the Transformer backbone with a causal TCN and conducted experiments. Due to time constraints, we performed these evaluations on the ETTh2 and ETTm2 datasets. The results, summarized below, demonstrate that using TCN as the backbone significantly improves performance over random initialization, highlighting the effectiveness of our self-supervised pretraining framework. While models with a Transformer backbone exhibit slightly better performance, the gap between the two is minimal, showcasing the robustness of our framework across different backbone architectures.
> >
> > | Models | Metrics | TCN MSE | TCN MAE | TCN Random init MSE | TCN Random init MAE | Transformer MSE | Transformer MAE |
> > |--------|---------|---------|---------|----------------------|----------------------|-----------------|-----------------|
> > | ETTh2  | 96      | **0.285** | **0.346** | 0.287                | 0.349                | 0.283           | 0.340           |
> > |        | 192     | **0.342** | **0.388** | 0.352                | 0.393                | 0.345           | 0.382           |
> > |        | 336     | **0.368** | **0.413** | 0.380                | 0.423                | 0.365           | 0.399           |
> > |        | 720     | **0.400** | **0.435** | 0.407                | 0.445                | 0.390           | 0.425           |
> > |        | Avg.    | **0.349** | **0.396** | 0.357                | 0.403                | 0.346           | 0.387           |
> > | ETTm2  | 96      | **0.172** | **0.265** | 0.176                | 0.272                | 0.165           | 0.256           |
> > |        | 192     | **0.226** | **0.299** | 0.238                | 0.304                | 0.221           | 0.294           |
> > |        | 336     | **0.282** | **0.337** | 0.284                | 0.335                | 0.279           | 0.330           |
> > |        | 720     | **0.372** | **0.390** | 0.378                | 0.392                | 0.364           | 0.385           |
> > |        | Avg.    | **0.263** | **0.323** | 0.269                | 0.326                | 0.257           | 0.316           |

---

> > > ### Author Response · Authors · 2024-11-19
> > >
> > > **Q4:** The author's core motivation remains to capture both global and local dependencies, which is similar to most existing works [1] [4]. In other words, this paper lacks a central challenging problem, making the contribution at the motivational level somewhat limited.
> > >
> > > **A4:** We sincerely thank the reviewer for their thoughtful comment regarding the central motivation and contribution of our work.
> > > We acknowledge that our motivation may not have been clearly expressed in the original submission. While capturing both global and local dependencies is indeed a common goal in time series modeling, our core motivation goes beyond this. Specifically, we aim to combine two powerful generative paradigms—the Diffusion Model and Auto-Regressive Pretraining—within a self-supervised framework to optimize time series representation learning.
> > > The novelty of our approach lies in this integration. The Diffusion Model excels at modeling intra-patch dependencies, capturing fine-grained local patterns within smaller time windows (or patches). Meanwhile, the Auto-Regressive mechanism focuses on inter-patch dependencies, effectively modeling sequential relationships between patches to capture broader, global patterns across time series data. This patch-wise combination of Diffusion and Auto-Regressive mechanisms is both novel and unique, enabling us to jointly model local and global dependencies in a way that has not been explored before.
> > > By leveraging this integrated framework, we address a central challenge: optimizing time series representation learning through the synergy of two generative models. Our results demonstrate significant improvements over existing methods, validating the effectiveness and innovation of our approach.
> > >
> > > **Q5:** Considering that the core motivation of this paper is to capture global and local dependencies, I suggest the author evaluate the model's performance on datasets with stable patterns, such as PEMS04. This is because datasets like ETT and Exchange have inherent issues with distributional shifts [5].
> > >
> > > **A5:** We sincerely thank the reviewer for their thoughtful suggestion regarding the evaluation of our method on datasets like PEMS04.
> > > We would like to clarify that our selection of datasets aligns with the current state-of-the-art works such as PatchTST and SimMTM. These datasets, including ETT and Exchange, represent real-world time series scenarios and reflect the diverse patterns and challenges encountered in practical applications, such as distributional shifts. This makes them highly relevant for evaluating the general effectiveness of our proposed method.
> > > Regarding PEMS04, while it is a well-known benchmark, it is specifically designed for GCN-based models due to its graph structure features. Our framework, which combines auto-regressive mechanisms and diffusion models, is not tailored to exploit graph-based features and thus may not align with the core design principles of PEMS04. Given the differing objectives and architectures, we believe that directly applying our method to PEMS04 may not provide a meaningful or fair comparison.
> > > That said, we acknowledge the value of exploring different types of datasets and plan to investigate broader domains in future work to further assess the versatility of our approach.

---

> > > > ### Comment · Reviewer_YroL · 2024-11-25
> > > >
> > > > Thanks for your rebuttal. However, I still have some concerns:
> > > >
> > > > (1) Due to differences in parameter size and structure among various models, fine-tuning some key hyperparameters is crucial. For example, the input length in paper [1] is fixed at 96, which results in a 50% performance drop for PatchTST on the Traffic dataset. I don't think this is fair.
> > > >
> > > > (2) Although the ETT and Exchange datasets are widely used, they exhibit severe biases and distribution shifts, leading to simple models like DLinear [2] outperforming classical Transformers by a significant margin. If datasets like PEMS04 and PEMS08, which have less severe distribution shifts, are used, DLinear performs far worse than Informer [3]. Moreover, recent studies [1] have also started evaluating models on PEMS04 and PEMS08. I don't believe comparing performance on such datasets is unfair.
> > > >
> > > > [1] itransformer: Inverted transformers are effective for time series forecasting
> > > >
> > > > [2] Are transformers effective for time series forecasting?
> > > >
> > > > [3] Informer: Beyond efficient transformer for long sequence time-series forecasting

---

> ### Author Response · Authors · 2024-11-27
>
> Dear Reviewer:
>
> Let us now address the concerns you raised based on the above rebuttal:
>
> **Q1**: Due to differences in parameter size and structure among various models, fine-tuning some key hyperparameters is crucial. For example, the input length in paper [1] is fixed at 96, which results in a 50% performance drop for PatchTST on the Traffic dataset. I don't think this is fair.
>
> **A1**: Thank you for raising concerns regarding the fairness of baseline performance comparisons. We acknowledge that our initial rebuttal may not have fully addressed your question.
>
> In our first-round response, under A1, we mentioned, "To ensure a fair comparison, we maintained consistency in key hyperparameters such as the learning rate schedule and dropout rate across all models." There was a wording mistake here; parameters like dropout are not key hyperparameters but rather non-core parameters. We sincerely apologize for any misunderstanding this may have caused. Below, we will systematically outline the efforts we made to ensure fairness in our evaluations:
>
> We would like to clarify that our experimental setup was conducted with fairness as a priority. Specifically, we used a consistent look-back window length of 336 and evaluated prediction horizons of [96, 192, 336, 720] across all self-supervised baselines, ensuring absolute fairness in the comparison. Moreover, we adhered to the model-specific parameters (e.g., d_model, patch_size, etc.) provided in the official open-source implementations of other methods, without any modification. To ensure a fair evaluation, we standardized certain non-core parameters, such as the dropout rate, across all models. Fine-tuning these parameters would risk overfitting, which we aimed to avoid.
>
> To sum up, we are committed to conducting rigorous and unbiased evaluations of the different self-supervised approaches through careful standardization and consistent experimental setups, ensuring the fairness of our experiments.
>
> We appreciate your detailed feedback and hope this clarifies our rationale.
>
>
> **Q2**: Although the ETT and Exchange datasets are widely used, they exhibit severe biases and distribution shifts, leading to simple models like DLinear [2] outperforming classical Transformers by a significant margin. If datasets like PEMS04 and PEMS08, which have less severe distribution shifts, are used, DLinear performs far worse than Informer [3]. Moreover, recent studies [1] have also started evaluating models on PEMS04 and PEMS08. I don't believe comparing performance on such datasets is unfair.
>
> **A2**: Thank you for raising the issue of datasets. Due to time constraints, we have now conducted experiments on the PEMS04 and PEMS08 datasets. For the comparison, we selected the best-performing self-supervised methods, including PatchTST and SimMTM. We followed the commonly used settings for the PEMS datasets as outlined in iTransformer, specifically using a look-back window of 96 and prediction horizons of [12, 24, 36, 48]. The experimental results are as follows:
>
> | Models | Metrics | TimeDART  |           | Random Init |       | SimMTM |       | PatchTST |       |
> | ------ | ------- | --------- | --------- | ----------- | ----- | ------ | ----- | -------- | ----- |
> |        |         | MSE       | MAE       | MSE         | MAE   | MSE    | MAE   | MSE      | MAE   |
> | PEMS04 | 12      | **0.087** | **0.197** | 0.092       | 0.201 | 0.100  | 0.205 | 0.098    | 0.205 |
> |        | 24      | **0.121** | **0.235** | 0.125       | 0.240 | 0.125  | 0.241 | 0.127    | 0.240 |
> |        | 36      | **0.149** | **0.260** | 0.156       | 0.269 | 0.157  | 0.271 | 0.163    | 0.278 |
> |        | 48      | **0.176** | **0.287** | 0.185       | 0.297 | 0.190  | 0.304 | 0.197    | 0.311 |
> |        | Avg.    | **0.133** | **0.245** | 0.140       | 0.252 | 0.143  | 0.255 | 0.146    | 0.259 |
> | PEMS08 | 12      | **0.109** | **0.223** | 0.111       | 0.225 | 0.110  | 0.223 | 0.122    | 0.230 |
> |        | 24      | **0.181** | **0.287** | 0.185       | 0.291 | 0.183  | 0.289 | 0.196    | 0.301 |
> |        | 36      | **0.239** | **0.296** | 0.247       | 0.306 | 0.246  | 0.305 | 0.280    | 0.321 |
> |        | 48      | **0.281** | **0.330** | 0.289       | 0.335 | 0.290  | 0.338 | 0.319    | 0.349 |
> |        | Avg.    | **0.203** | **0.284** | 0.208       | 0.289 | 0.207  | 0.289 | 0.229    | 0.300 |
>
> As shown in the table, TimeDART outperforms our baseline methods on these datasets, demonstrating its effectiveness. Our implement scripts can be found in https://anonymous.4open.science/r/PEMS-scripts-2024

---

### Official Review · Reviewer_YF6R · 2024-10-30

**Soundness:** 3
**Presentation:** 3
**Contribution:** 3
**Rating:** 8
**Confidence:** 3

**Summary:**

Recently, effectively capturing both the global sequence dependence and local detail features within time series data remains challenging. To address this, the authors propose a novel generative self-supervised method called TimeDART, denoting Diffusion Auto-regressive Transformer for time series forecasting.

**Strengths:**

1. We propose a novel generative self-supervised learning framework, TimeDART, which in tegrates diffusion and auto-regressive modeling to effectively learn both global sequence dependencies and local detail features from time series data, addressing the challenges of capturing comprehensive temporal characteristics.
 2. We design a cross-attention-based denoising decoder within the diffusion mechanism, which enables adjustable optimization difficulty during the self-supervised task. This design significantly enhances the model’s ability to capture localized intra-patch features, improving the effectiveness of pre-training for time series forecasting. Diffusion models and autoregressive attention mechanisms are rare collaborations in temporal tasks, and this field brings new ideas.
3. The experimental results show that the combination of Mamba and propagation mechanism is very effective, and it also exceeds the predictive performance of supervised learning.

**Weaknesses:**

1. The reviewer is concerned about the computational overhead associated with this approach. The reviewer's core concern stems from the introduction of diffusion mechanisms, which can lead to a substantial increase in training and inference overhead for the model as a whole. If the model is expensive, the computational conditions required to solve the real task will be severe. Therefore, the authors need to report the actual time of the training and inference phase of other baseline models in the future and report the GPU and server model.
2. The motivation for choosing to use a causal mechanism in Transformer requires further explanation. After all, time series data is encoded with more complex patterns. In particular, there are random changes caused by extreme weather events in the meteorological data, and this causal relationship is strong. But many things cause and effect is unclear, so whether such a component is appropriate needs to be used for the specific task.

**Questions:**

Please see weaknesses.

---

> ### Author Response · Authors · 2024-11-19
>
> **Q1:** The reviewer is concerned about the computational overhead associated with this approach. The reviewer's core concern stems from the introduction of diffusion mechanisms, which can lead to a substantial increase in training and inference overhead for the model as a whole. If the model is expensive, the computational conditions required to solve the real task will be severe. Therefore, the authors need to report the actual time of the training and inference phase of other baseline models in the future and report the GPU and server model.
>
> **A1:** We sincerely thank the reviewer for highlighting the concern regarding computational overhead. This is indeed a critical factor for the practical application of our method.
> To address this, we would like to clarify that all our experiments, including those involving the denoising diffusion mechanism, were conducted on a single NVIDIA RTX 4090 24GB GPU. This demonstrates that TimeDART can be efficiently trained and evaluated on high-end GPUs without excessive computational requirements.
> To further illustrate the computational feasibility, we have summarized the training and inference times, as well as memory usage(look-back window=336, predicted window=96), for TimeDART and the baseline models in the table below. These results show that TimeDART maintains competitive efficiency and scalability even for larger datasets like Traffic.
>
> **Traffic**
>
> | Methods                | Params                                | Training Time/per epoch                |
> |------------------------|---------------------------------------|----------------------------------------|
> | TimeDART               | 2.60M (pretraining) / 1.11M (finetuning) | 510s (pretraining) / 349s (finetuning) |
> | SimMTM                 | 942.51M (pretraining) / 4.53M (finetuning) | 5540min (pretraining) / 1183min (finetuning) |
> | PatchTST (supervised)  | 64.32M                                | 672min                                 |
> | TimeMAE                | 940.62K (pretraining) / 1.20M (finetuning) | 90min (pretraining) / 152s (finetuning) |
>
> **ETTh1**
>
> | Methods                | Params                                  | Training Time/per epoch               |
> |------------------------|-----------------------------------------|---------------------------------------|
> | TimeDART               | 1.85M (pretraining) / 541.86K (finetuning) | 18s (pretraining) / 20s (finetuning)  |
> | SimMTM                 | 62.14M (pretraining) / 1.05M (finetuning) | 71min (pretraining) / 16min (finetuning) |
> | PatchTST (supervised)  | 4.27M                                   | 129s                                  |
> | TimeMAE                | 67.53K (pretraining) / 193.59K (finetuning) | 72s (pretraining) / 11s (finetuning)  |

---

> ### Author Response · Authors · 2024-11-19
>
> **Q2:** The motivation for choosing to use a causal mechanism in Transformer requires further explanation. After all, time series data is encoded with more complex patterns. In particular, there are random changes caused by extreme weather events in the meteorological data, and this causal relationship is strong. But many things cause and effect is unclear, so whether such a component is appropriate needs to be used for the specific task.
>
> **A2:** We sincerely thank the reviewer for raising this insightful question regarding the use of a causal mechanism in the Transformer. This is indeed a critical component of our approach, and we appreciate the opportunity to clarify its role and motivation.
> Regarding the use of a causal mechanism in the Transformer, I would like to clarify that the Causal Mask is only employed during the pre-training phase of the model’s representation network. In the downstream tasks, we align with other baselines by removing the causal mask and using full attention instead.
> The motivation for introducing the causal mechanism during pre-training has two main aspects:
> 1. **Mechanistic Consideration**: The causal mechanism ensures that the auto-regressive property is properly implemented during pre-training. Without it, there would be a risk of information leakage, which would compromise the training process.
> 2. **Motivational Consideration**: Forecasting tasks inherently involve causal relationships, and introducing the causal mechanism during pre-training allows the representation network to naturally learn such causal dependencies. From our ablation experiments, we found that the performance improvement with the causal mask is significant compared to not using it.
> We appreciate your point about the ambiguity of causal relationships in some time series data. Indeed, some sequences, such as those involving random changes (e.g., weather events), may not exhibit clear causal patterns. To address this, we removed the causal mask in the downstream tasks to maintain consistency with the baselines, which in turn encourages the model to explore non-causal dependencies in the data. This approach has proven effective in improving performance across a broader range of forecasting tasks. In future work, we will further explore the role of non-causal dependencies in time series forecasting and evaluate how different types of dependencies impact the model's performance.
>
> We appreciate your point about the ambiguity of causal relationships in some time series data. Indeed, some sequences, such as those involving random changes (e.g., weather events), may not exhibit clear causal patterns. To address this, we removed the causal mask in the downstream tasks to maintain consistency with the baselines, which in turn encourages the model to explore non-causal dependencies in the data. This approach has proven effective in improving performance across a broader range of forecasting tasks. In future work, we will further explore the role of non-causal dependencies in time series forecasting and evaluate how different types of dependencies impact the model's performance.

---

> > ### Comment · Reviewer_YF6R · 2024-11-19
> > **Feedback to Authors**
> >
> > Dear Authors,
> >
> > I have no further concerns. In addition, for A2, the authors' carefully designed experiments can validate of some claim, and if the 'proven' is used it must be to give a formal analysis.
> >
> > Best Regards,
> > Reviewer

---

> > > ### Author Response · Authors · 2024-11-21
> > >
> > > Dear Reviewer,
> > >
> > > Thank you for taking the time to carefully read my responses and for providing such thoughtful and constructive feedback. I sincerely appreciate your suggestion regarding the use of precise terminology like "proven" and will take great care with my wording and expressions in future work.
> > >
> > > I am especially grateful for your acceptance score. This score is very important to us and has greatly motivated me to further explore and contribute to this field in the future.
> > >
> > > Thank you once again for your valuable comments and evaluation, which are deeply appreciated.
> > >
> > > Best regards, Authors

---

### Official Review · Reviewer_J3uD · 2024-11-10

**Soundness:** 3
**Presentation:** 3
**Contribution:** 3
**Rating:** 5
**Confidence:** 4

**Summary:**

The paper introduces TimeDART (Diffusion Auto-Regressive Transformer for Time Series Forecasting), a novel self-supervised learning framework designed to enhance time series forecasting. TimeDART addresses key challenges in the field, particularly capturing both long-term dependencies and local features within time series data.

**Strengths:**

+ Combining self-attention for inter-patch dependencies, diffusion mechanisms for intra-patch dependencies, and auto-regressive optimization is an interesting and innovative approach to time series forecasting.
+ The diffusion-based reverse process for reconstructing the sequence is novel in the context of time series forecasting.
+ Good writing and organization.

**Weaknesses:**

+ The combination of Transformer-based attention mechanisms with the denoising diffusion process introduces substantial computational overhead. The paper mentions running experiments on a single NVIDIA RTX 4090 GPU, but it doesn't provide detailed insights into the time and memory consumption required for training. How scalable is TimeDART for larger datasets or real-time applications?
+ The experiments conducted on noise scheduling, the number of diffusion steps, and the number of layers in the denoising network highlight a significant degree of sensitivity to hyperparameter selection. How the model will perform well in practical settings with minimal tuning?
+ The cross-domain evaluation shows strong results on energy datasets but weaker performance on datasets like Exchange. Is TimeDART overly sensitive to the type of data it is pre-trained on? For instance, does it struggle with financial or highly volatile datasets because of the lack of shared characteristics between domains (e.g., energy vs. finance)? Could this method benefit from domain adaptation techniques to make the cross-domain transfer more robust?
+ The model uses a denoising diffusion loss, which may not be the most suitable for every type of forecasting task. How does TimeDART perform with other loss functions (e.g., Quantile Loss, Huber Loss) that are often used in time series forecasting tasks where the goal is to forecast confidence intervals or robustly handle outliers?

**Questions:**

Check Weaknesses.

---

> ### Author Response · Authors · 2024-11-19
>
> **Q1:** The combination of Transformer-based attention mechanisms with the denoising diffusion process introduces substantial computational overhead. The paper mentions running experiments on a single NVIDIA RTX 4090 GPU, but it doesn't provide detailed insights into the time and memory consumption required for training. How scalable is TimeDART for larger datasets or real-time applications?
>
> **A1:**  We sincerely appreciate the reviewer’s insightful comments regarding the computational overhead. To clarify, the denoising diffusion process in our framework is only used during pretraining, while the downstream tasks adopt a lightweight linear layer for single-step predictions, significantly reducing inference complexity.
> To address scalability concerns, we conducted experiments on runtime and memory usage (look-back window=336, predicted window=96). Results, including the largest Traffic dataset, demonstrate that our model requires less time and memory compared to other methods. Detailed results are presented in the table below.
>
> **Traffic**
>
> | Methods                | Params                                | Training Time/per epoch                |
> |------------------------|---------------------------------------|----------------------------------------|
> | TimeDART               | 2.60M (pretraining) / 1.11M (finetuning) | 510s (pretraining) / 349s (finetuning) |
> | SimMTM                 | 942.51M (pretraining) / 4.53M (finetuning) | 5540min (pretraining) / 1183min (finetuning) |
> | PatchTST (supervised)  | 64.32M                                | 672min                                 |
> | TimeMAE                | 940.62K (pretraining) / 1.20M (finetuning) | 90min (pretraining) / 152s (finetuning) |
>
> **ETTh1**
>
> | Methods                | Params                                  | Training Time/per epoch               |
> |------------------------|-----------------------------------------|---------------------------------------|
> | TimeDART               | 1.85M (pretraining) / 541.86K (finetuning) | 18s (pretraining) / 20s (finetuning)  |
> | SimMTM                 | 62.14M (pretraining) / 1.05M (finetuning) | 71min (pretraining) / 16min (finetuning) |
> | PatchTST (supervised)  | 4.27M                                   | 129s                                  |
> | TimeMAE                | 67.53K (pretraining) / 193.59K (finetuning) | 72s (pretraining) / 11s (finetuning)  |
>
>
> **Q2:** The experiments conducted on noise scheduling, the number of diffusion steps, and the number of layers in the denoising network highlight a significant degree of sensitivity to hyperparameter selection. How the model will perform well in practical settings with minimal tuning?
>
> **A2:** We appreciate the reviewer’s valuable comment on hyperparameter sensitivity. Indeed, hyperparameter tuning can be time-consuming, and we have devoted significant effort to exploring this aspect. Through extensive experiments, we identified a fundamental guideline: ensure a smooth noise addition process and maintain a balance between the denoising network and the representation network’s parameter scales.
> Specifically:
> - **Total Noise Steps:** We found that performance differences across values (e.g., 750, 1000, 1250) are relatively minor. The commonly used 1000-step setting strikes a good balance between performance and training efficiency, requiring no significant adjustment.
> - **Noise Scheduler:** Our experiments confirm that the cosine scheduler consistently outperforms the linear scheduler. As such, we recommend the cosine scheduler as a robust default.
> - **Denoising Network Layers:** Using overly deep networks can adversely affect representation learning. Through experimentation, we determined that a one-layer denoising network is optimal, providing an effective trade-off between model capacity and computational efficiency.

---

> ### Author Response · Authors · 2024-11-19
>
> **Q3:** The cross-domain evaluation shows strong results on energy datasets but weaker performance on datasets like Exchange.
> Is TimeDART overly sensitive to the type of data it is pre-trained on? For instance, does it struggle with financial or highly volatile datasets because of the lack of shared characteristics between domains (e.g., energy vs. finance)?
>  Could this method benefit from domain adaptation techniques to make the cross-domain transfer more robust?
>
> **A3:** We sincerely thank the reviewer for raising this important and insightful question. Our response is structured as follows:
> - Clarification on Cross-Domain Experiments
> We respectfully clarify that our original submission did not include cross-domain evaluations on the Exchange dataset. At the time of submission, these experiments had not yet been conducted. Therefore, the observed results on Exchange reflect in-domain performance rather than cross-domain evaluation. We appreciate the opportunity to clarify this point and acknowledge that a more comprehensive cross-domain analysis is valuable.
> - New Cross-Domain Experiments and Model Robustness
>   To address your concerns regarding model sensitivity and the generalizability of cross-domain experiments, we worked tirelessly to design and conduct additional evaluations. Specifically, we performed a general pretraining on all eight datasets and fine-tuned TimeDART on four datasets: ETTh2, ETTm2, Exchange, and Electricity. We selected UniTime, GPT4TS, and PatchTST as baselines, ensuring alignment with UniTime’s experimental setup (look-back window of 96, predicted windows of {96, 192, 336, 720}). These settings allowed for a fair and consistent comparison across models.
>   Due to time constraints, we have so far focused on these four datasets but plan to extend our evaluations to additional datasets to thoroughly assess the model’s cross-domain performance. The new results, summarized in the **table1** below, highlight TimeDART's robustness and potential for further improvement through domain adaptation techniques.
>
> **Q4:** The model uses a denoising diffusion loss, which may not be the most suitable for every type of forecasting task. How does TimeDART perform with other loss functions (e.g., Quantile Loss, Huber Loss) that are often used in time series forecasting tasks where the goal is to forecast confidence intervals or robustly handle outliers?
>
> **A4:** We sincerely thank the reviewer for raising this insightful question. Our response is structured as follows:
> - Clarification of Diffusion Loss Usage
> We would like to clarify that the diffusion loss is used exclusively during the self-supervised pretraining phase, not in downstream tasks. Its purpose is to model the underlying data distribution via a denoising process, which helps TimeDART capture richer representations. While the diffusion loss has a form similar to MSE, it is fundamentally different from traditional forecasting losses like Quantile Loss and Huber Loss. Unlike these losses, which extend MSE or MAE for specific objectives, the diffusion loss is designed to improve the model’s ability to generate diverse, high-quality predictions for complex forecasting tasks.
> - MSE in Downstream Tasks
> In downstream forecasting tasks, we follow the standard practice of using MSE as the optimization objective, which aligns with the majority of self-supervised methods. To ensure a fair comparison, all baselines in our experiments also use MSE as their loss function. This allows us to isolate and evaluate the benefits of the pretrained representations.
> We acknowledge the reviewer’s suggestion to explore alternative loss functions, such as Quantile Loss and Huber Loss, for downstream tasks. This is indeed a valuable direction, and we plan to extend our experiments in future work to assess the impact of these losses on forecasting performance.

---

> ### Author Response · Authors · 2024-11-19
> **Table1: Comparison on cross-domain setting. Model is trained on 8 datasets and finetuned on a single dataset.**
>
> | Models             | Metrics | TimeDART (MSE) | TimeDART (MAE) | Random init (MSE) | Random init (MAE) | Unitime (MSE) | Unitime (MAE) | GPT4TS (MSE) | GPT4TS (MAE) | PatchTST (MSE) | PatchTST (MAE) |
> | ------------------ | ------- | -------------- | -------------- | ----------------- | ----------------- | :------------ | ------------- | ------------ | ------------ | -------------- | -------------- |
> | ALL -> ETTh2       | 96      | **0.293**      | **0.339**      | 0.296             | 0.342             | 0.296         | 0.345         | 0.303        | 0.349        | 0.314          | 0.361          |
> |                    | 192     | **0.374**      | **0.390**      | 0.382             | 0.394             | **0.374**     | 0.394         | 0.391        | 0.399        | 0.407          | 0.411          |
> |                    | 336     | **0.410**      | **0.419**      | 0.422             | 0.430             | 0.415         | 0.427         | 0.429        | 0.449        | 0.437          | 0.443          |
> |                    | 720     | **0.425**      | **0.444**      | 0.436             | 0.449             | **0.425**     | **0.444**     | 0.430        | 0.449        | 0.434          | 0.448          |
> |                    | Avg.    | **0.376**      | **0.398**      | 0.384             | 0.404             | 0.378         | 0.403         | 0.386        | 0.406        | 0.398          | 0.416          |
> | ALL -> ETTm2       | 96      | **0.180**      | 0.270          | 0.195             | 0.289             | 0.183         | **0.266**     | 0.229        | 0.304        | 0.240          | 0.318          |
> |                    | 192     | **0.245**      | **0.305**      | 0.267             | 0.333             | 0.251         | 0.310         | 0.287        | 0.338        | 0.301          | 0.352          |
> |                    | 336     | **0.308**      | **0.348**      | 0.311             | 0.355             | 0.319         | 0.351         | 0.337        | 0.367        | 0.367          | 0.391          |
> |                    | 720     | **0.413**      | **0.409**      | 0.431             | 0.421             | 0.420         | 0.410         | 0.430        | 0.416        | 0.451          | 0.432          |
> |                    | Avg.    | **0.287**      | **0.333**      | 0.301             | 0.350             | 0.293         | 0.334         | 0.321        | 0.356        | 0.340          | 0.373          |
> | ALL -> Exchange    | 96      | **0.082**      | 0.211          | 0.094             | 0.212             | 0.086         | **0.209**     | 0.142        | 0.261        | 0.137          | 0.260          |
> |                    | 192     | **0.172**      | **0.299**      | 0.212             | 0.332             | 0.174         | **0.299**     | 0.224        | 0.339        | 0.222          | 0.341          |
> |                    | 336     | 0.329          | 0.418          | 0.365             | 0.442             | **0.319**     | **0.408**     | 0.377        | 0.448        | 0.372          | 0.447          |
> |                    | 720     | **0.861**      | **0.697**      | 0.886             | 0.709             | 0.875         | 0.701         | 0.939        | 0.736        | 0.912          | 0.727          |
> |                    | Avg.    | **0.361**      | 0.406          | 0.389             | 0.424             | 0.364         | **0.404**     | 0.421        | 0.446        | 0.411          | 0.444          |
> | ALL -> Electricity | 96      | **0.178**      | **0.269**      | 0.189             | 0.287             | 0.189         | 0.287         | 0.198        | 0.290        | 0.202          | 0.293          |
> |                    | 192     | **0.182**      | **0.273**      | 0.200             | 0.290             | 0.199         | 0.291         | 0.234        | 0.325        | 0.223          | 0.318          |
> |                    | 336     | **0.199**      | **0.297**      | 0.206             | 0.301             | 0.214         | 0.305         | 0.249        | 0.338        | 0.223          | 0.318          |
> |                    | 720     | **0.241**      | **0.332**      | 0.251             | 0.333             | 0.254         | 0.335         | 0.289        | 0.366        | 0.259          | 0.341          |
> |                    | Avg.    | **0.200**      | **0.293**      | 0.212             | 0.305             | 0.216         | 0.305         | 0.251        | 0.338        | 0.221          | 0.311          |

---

> > ### Comment · Reviewer_J3uD · 2024-11-26
> >
> > Thank you for the explanation and further experiments. Overall, the paper is good and the ideas are novel. My only concern is the experimental setup. As other reviewers have said, some unfairness keeps the score at 5.

---

> ### Author Response · Authors · 2024-11-27
>
> Dear Reviewer:
>
> Thank you for your valuable feedback. We would like to clarify that our experimental setup was conducted with fairness as a priority. Specifically, following the practices of other works such as PatchTST and SimMTM, we used a consistent look-back window length of 336 and evaluated prediction horizons of [96, 192, 336, 720] across all self-supervised baselines, ensuring absolute fairness in the comparison. Moreover, we adhered to the model-specific parameters (e.g., d_model, patch_size, etc.) provided in the official open-source implementations of other methods, without any modification. To ensure a fair evaluation, we standardized certain non-core parameters, such as the dropout rate, across all models. Fine-tuning these parameters would risk overfitting, which we aimed to avoid.
>
> To sum up, we are committed to conducting rigorous and unbiased evaluations of the different self-supervised approaches through careful standardization and consistent experimental setups, ensuring the fairness of our experiments.

---

### Meta-Review · Area_Chair_n6eA · 2024-12-15

**Metareview:**

The paper proposes TimeDART, a novel self-supervised learning framework that integrates diffusion mechanisms and autoregressive Transformers to capture global dependencies and local detail features in time series forecasting. Strengths include its innovative combination of diffusion and autoregressive techniques, a cross-attention denoising decoder for effective self-supervised pretraining, and extensive experiments showing competitive performance in forecasting tasks. However, weaknesses include significant computational overhead due to the diffusion process, limited exploration of cross-domain generalization and inter-channel dependencies, sensitivity to hyperparameter tuning, and less-than-expected improvements over state-of-the-art methods. Additionally, the paper lacks evaluation on broader tasks like classification and more diverse baselines. These limitations suggest the contributions, while promising, may not yet be sufficient to justify acceptance, leading to a recommendation for rejection.

**Additional Comments On Reviewer Discussion:**

During the rebuttal period, reviewers raised concerns about TimeDART’s computational overhead, limited cross-domain generalization, sensitivity to hyperparameters, insufficient evaluation against competitive baselines, and its lack of metrics addressing generative capabilities. The authors responded by clarifying the use of computational resources, conducting additional experiments on cross-domain datasets, and providing comparisons with more baselines such as UniTime. They also explained the architectural differences limiting direct comparisons with fully generative diffusion models. Despite these efforts, key concerns about the limited performance improvements in cross-domain settings, unresolved issues with generative evaluation, and reliance on high computational resources persisted. These unresolved points weighed heavily in the final decision to reject, as the improvements during the rebuttal did not sufficiently address the core limitations identified by reviewers.

---

### Decision · Program_Chairs · 2025-01-22

Reject